# GeoBench: A new benchmark on Symbolic Regression with Geometric Expressions

## Abstract

Symbolic regression (SR) is a highly effective approach for discovering mathematical expressions directly from data. With the proliferation of various SR methods, SRBench (La Cava et al., 2021) has made an important contribution by offering a standardized evaluation framework that includes 130 SR datasets and assesses 14 SR methods. Nevertheless, the methods incorporated in SRBench are somewhat outdated, and the benchmark dataset does not encompass results from more recent approaches, such as SNIP (Meidani et al., 2024). Furthermore, the evaluation metrics employed in SRBench fail to fully capture the breadth of symbolic regression capabilities, and the benchmark data itself exhibits scientific inconsistencies. Although Matsubara et al. (2022) address some of these issues, their approach remains incomplete. In response, we propose a novel benchmark consisting of 71 expressions derived from geometric contexts, which are categorized into three difficulty levels: easy, medium, and hard. We conduct an evaluation of 20 SR methods on these expressions, focusing exclusively on the symbolic regression capabilities of each model. These capabilities are measured in terms of recovery rates across the different difficulty levels and in aggregate. Our study provides a comprehensive methodology for reproducing the experiments and includes results for newly developed SR methods using this updated benchmark. The findings reveal significant variability in the symbolic regression performance across the evaluated models.

## 1 Instruction

In many aspects of life, various phenomena can be described by mathematical equations, such as Newton's Second Law and the law of gravity. Symbolic regression (SR) is a powerful tool for uncovering these underlying relationships. Specifically, SR seeks to discover a mathematical expression that links input and output data. Unlike traditional machine learning techniques, such as neural networks, SR might offers greater interpretability and superior generalization, avoiding the complexity often associated with opaque models. For instance, the movement function of a pendulum is simpler and more effective than the matrix values of an MLP. Due to these advantages, symbolic regression has been applied across diverse fields, including physics (Sun et al., 2021; Udrescu & Tegmark, 2020; Schmidt & Lipson, 2009), network control (Sharan et al., 2022), finance (La Malfa et al., 2021), and material science (Wang et al., 2019).

However, symbolic regression (SR) poses significant challenges due to its expansive search space. The inclusion of constants further complicates the task, as they increase the complexity of potential solutions. In fact, SR has been formally proven to be an NP-complete problem (Virgolin & Pissis, 2022; Song et al., 2024).

As background for some symbolic regression methods, expression trees became a popular approach for tackling SR tasks. An expression tree consists of *internal nodes* representing mathematical operators (e.g., $+, -, \times, \div, \log, \exp, \sin, \cos$) and *leaf nodes* that represent constants (e.g., $1, 2$) or variables (e.g., $x$). By recursively evaluating sub-trees, the expression tree generates a corresponding mathematical expression. The construction of an expression tree typically follows a recursive method, where operators are added in pre-order traversal until no further additions are possible. This transforms SR into a sequence generation problem, akin to tasks in natural language processing (NLP).

Symbolic regression involving constants presents a particularly challenging task. Linear methods often treat constants within equations as parameters in a linear regression framework, allowing them to be estimated by solving the associated function. For evolution models, models typically sample random values to substitute for constants in the generated expressions. If an expression yields a high loss value, the corresponding random constant is discarded, while constants producing lower losses are retained. In expression tree models, constants are preserved by introducing a constant token. This allows the model to solve for the constant value during the error calculation phase. At this stage, the constants are treated as input variables, and the error is treated as the target value. Optimization algorithms, such as BFGS (Roger Fletcher & Sons, 2013), are commonly employed to fine-tune the constants and minimize error.

With the development of SR methods came the emergence of benchmarks. Nyugen (Uy et al., 2011) introduced one of the early symbolic regression benchmarks, comprising 12 short expressions designed to evaluate SR techniques across a range of simple to moderately complex equations. Similarly, Jin (Jin et al., 2019), Neat, and Keijzer (Keijzer, 2003) created their own datasets to test the symbolic regression abilities of their models. However, the data in these benchmarks have limitations, as they often involve non-elementary functions, such as the expression $\sum_{i=1}^{x_1} \frac{1}{i}$. Consequently, these datasets are sometimes more suited to assessing curve-fitting abilities rather than the capacity to discover underlying symbolic functions.

As well, some benchmarks have been developed to assess specific aspects of symbolic regression capabilities. For example, Nyugen$^c$ primarily evaluates SR models' ability to handle constants, as all its equations include constant parameters. The R rationals and R* benchmarks (McDermott et al., 2012) are designed to test models' abilities to solve complex fractional equations, while the Livermore benchmark (Mundhenk et al., 2021b) focuses on equations containing $\cos, \sin, \log, \exp,$ and power functions. However, these benchmarks often lack real-world applicability, as expressions like $\log(x_1 + 1) + \log(x_1^2 + x_1) + \log(x_1)$ are primarily suited for testing, not use in real world . As well, these benchmarks has no uniform metric since some models tests for $R^2$ and the other uses it for symbolic recovery rate.

The introduction of the AIFeynman dataset (Udrescu & Tegmark, 2020) marked a significant step forward in SR benchmarking. This dataset comprises 100 equations derived from the Feynman Lectures on Physics (Feynman et al., 2015) with 20 complement as bonus equations and serves as a robust benchmark for SR tasks. Recently, the SRBench team (La Cava et al., 2021) has combined 118 equations from this dataset with the Strogatz dataset (Strogatz, 2018), which includes 14 equations modeling nonlinear and chaotic dynamical processes, providing a more comprehensive evaluation of SR methods. Additionally, SRBench includes different noise levels ranging from $0.0, 0.1, 0.01, 0.001$, to assess the models' ability to handle noisy data. They also utilize real-world datasets to assess the machine learning capabilities of symbolic models, which falls outside the scope of this paper. This benchmark evaluates 14 SR methods as baselines, resolving issues seen in earlier datasets, such as a focus on equations primarily suited for testing and not for real-world scientific experiments. It also provides a framework to reproduce results and test new models. However, SRBench has some limitations, as more than half of the evaluated methods are based on genetic programming (GP) approaches, and many are from before 2022. Furthermore, the benchmark includes some scientifically unrealistic assumptions, and has been criticized for its oversimplified sampling process and inappropriate formulas (Matsubara et al., 2022).

Subsequently, Matsubara et al. (2022) attempted to address some of the existing issues; however, their evaluation was restricted to only six methods. This limited selection of baselines may result in an insufficient comparison, particularly when assessing the performance of new models introduced into the field.

To address these issues, we propose our geometric dataset. It consists of 2D and 3D geometric problems, such as calculating the area of a triangle given the lengths of its three sides. These problems are meaningful in real-world applications and complement existing physical symbolic regression datasets. We categorize these datasets into three difficulty levels: easy, medium, and hard. We evaluate our 71 datasets using 20 SR baselines from 8 different approaches. The metrics of our benchmark are twofold: (1) the symbolic recovery rate across each difficulty level and overall, and (2) the number of expressions that can be discovered when models are allowed to run for 100 parallels.

## 2 DIFFERENT APPROACH OF SYMBOLIC REGRESSION

**Linear Methods:** The SINDy method (Kaiser et al., 2018) applies the L1 Loss to reduce the number of active basis functions in a linear regression framework, thereby distilling a simple equation as a linear combination of candidate terms from a predefined library. Although SINDy is known for its interpretability and speed, its performance heavily depends on the selection of the predefined library. If the true solution is not a combination of terms in the library, SINDy is unable to identify it. Recently, the KAN model (Liu et al., 2024) has emerged using spline methods as an alternative to improve upon these limitations.

**Genetic Programming:** The genetic programming method (Schmidt & Lipson, 2009; Augusto & Barbosa, 2000; Gustafson et al., 2005) represents expressions as trees, which serve as the populations in the algorithm. Mutation and crossover operations modify the trees by changing sub-trees or exchanging parts of the tree. The advantage of genetic algorithms in symbolic learning is their ability to iteratively modify the expression tree via genetic recombination, enabling the model to explore a wide range of expressions. However, a significant disadvantage is the tendency of genetic algorithms to overfit; once the algorithm veers toward an incorrect solution, it is often difficult to recover a correct path to the truth.

**Deep Learning Methods:** There are two main approaches to using deep learning in symbolic regression. One approach leverages neural networks to identify relationships between variables and merge them to reduce the search space (Udrescu & Tegmark, 2020; Udrescu et al., 2020). While this method simplifies the search, it requires large amounts of data and does not always succeed in fitting the correct equations. The other approach replaces traditional network components (e.g., linear layers or activation functions) with symbolic functions and applies L1 loss to reduce active modules, thus simplifying the output (Martius & Lampert, 2016; Sahoo et al., 2018). This approach achieves lower MSE, but optimizing the sparse network to precisely recover the correct equation is extremely challenging.

**Deep Reinforcement Learning Methods:** The deep reinforcement learning approach (Petersen et al., 2019) frames symbolic regression as a sequential decision-making problem, where models take actions at each step (e.g., adding or modifying terms) based on the current state, which is evaluated using a recurrent neural network (or LSTM). After each generation, the models learn from the best-generated expressions, guided by a reward function. This method effectively narrows the search space but can suffer from overfitting and lack of exploration.

**Traditional Machine Learning Methods:** This approach (Sun et al., 2022; Xu et al., 2024) is similar to deep reinforcement learning but uses Monte Carlo Tree Search (MCTS) instead of neural networks to guide the search process. By avoiding the need for neural network training, this method is faster for smaller problems but struggles with more complex equations.

**Transformer-Based Pretrain Methods:** Inspired by the GPT models (Radford et al., 2018), transformer-based symbolic regression models (Kamienny et al., 2022) pretrain on large sets of artificial expressions and use this pretraining to generate expressions from input data. Subsequently, genetic programming or reinforcement learning (Holt et al., 2022; Landajuela et al., 2022) is employed to refine the output of the transformer models. While transformers provide excellent initial solutions, they may struggle with out-of-distribution data, leading to overfitting or poor performance on unseen tasks.

**Bayesian Methods:** Bayesian symbolic regression (Jin et al., 2019; Guimerà et al., 2020) leverages prior knowledge (e.g., preferences for basis functions, operators, or original features) and produces symbolic expressions as a linear combination of concise terms, controlled by a prior distribution. The symbolic regression problem is solved by sampling expression trees from the posterior distribution using a Markov Chain Monte Carlo (MCMC) algorithm. Although this method conserves memory, it can be computationally expensive and may struggle to produce accurate results due to the limitations of MCMC sampling.

**Brute-Force Search Methods:** Given that symbolic regression seeks simple expressions to describe phenomena, the true expression trees often have limited depth (e.g., maximum 6 layers). This observation motivates brute-force methods, which enumerate possible expressions layer by layer (Ruan et al., 2024), as the $n + 1$-th layer can be constructed by combining elements from the $n$-th layer. GPU-based implementations can accelerate this search process, making brute-force methods effec-

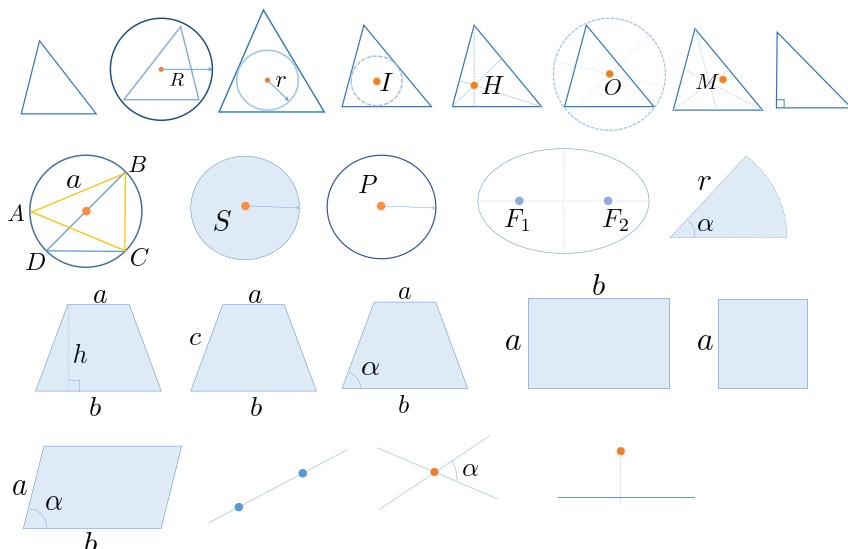

Figure 1: the 2D geometric objects in our dataset including triangles, circles, trapezoids, elliptic, squares, rectangles, lines and point.

tive for finding simple expressions with few variables, although they struggle with larger and more complex problems because of GPU's memory.

## 3 GEOMETRIC DATASET

### 3.1 DATASETS

Diving into the details of our geometry dataset, it's divided into two main sections: 2-D and 3-D geometry. The first section is a thorough compilation of 2-D geometrical shapes such as triangles, rectangles, squares, and circles, complete with their corresponding equations. In the second section, the dataset expands into the realm of 3-D geometry, presenting a wide array of shapes including vectors, spheres, various solids, and pyramids, each paired with their relevant equations.

***2-D part:*** The dataset begins with various types of triangles. We assess the ability to determine the perimeter and area of triangles given different sets of known values: three sides (SSS), two sides with the included angle (SAS), and two angles with the included side (AAS) or the opposite sides (ASA). These four methods constitute the foundational techniques for establishing triangle congruence and equality.

For right-angled triangles, the dataset facilitates the calculation of the perimeter and area using the lengths of the right sides and the hypotenuse, or by employing the length of one right side and the angle opposite to it.

Incorporated into this dataset are three pivotal laws of trigonometry: the Cosine Theorem (Law of Cosines), the Pythagorean Theorem, and the Sine Theorem (Law of Sines). Utilization of these theorems allows for the resolution of the perimeter and area for a variety of straightforward geometrical constructs.

Moreover, the dataset tackles more challenging computations such as determining the circumcircle and incircle radix of a triangle based solely on its three side lengths.

Expanding beyond simple measurements, we also delve into coordinate geometry. The dataset includes the calculation of the horizontal coordinates for four significant points within a triangle: the centroid (center of mass), the incenter (intersection of angle bisectors), the circumcenter (intersection of perpendicular bisectors), and the orthocenter (intersection of altitudes). These calculations are vital for a deeper understanding of a triangle's geometric properties and their applications.

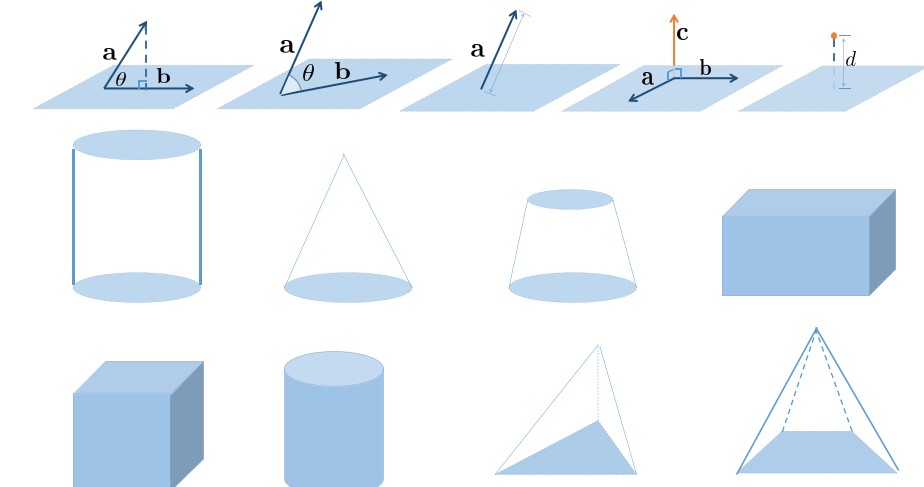

Figure 2: the 3D geometric objects in our dataset including three-dimensional vectors, cylinder, cones, frustums, sphere, cuboids, cubes, pyramids and tetrahedrons.

Venturing beyond triangular shapes, our collection encompasses trapezoids, specifically focusing on isosceles trapezoids. By utilizing the dimensions of the upper and lower bases, height, sides, or the angles adjacent to the base, one can deduce both the perimeter and area of these quadrilaterals.

The dataset also embraces the circular and elliptical geometries. It allows for the calculation of a circle's perimeter (or circumference) using its radius, as well as the perimeter and area of a sector by its central angle and radius. For ellipses, the major and minor axes serve as the basis for determining the area and locating the focal points.

Additionally, the dataset includes rectangles and squares. Given the lengths of their edges, we can easily determine their perimeter and area.

Lastly, the dataset serves as a resource for analytical geometry concerning lines and points. It enables the determination of the horizontal and vertical coordinates where two lines intersect, based on their slopes and intercepts. It further aids in calculating the slope and intercept of a line passing through two points, given their horizontal and vertical coordinates. Additionally, it provides the tools to find the directed distance from a point to a line, integrating the line's slope and intercept with the point's coordinates.

***3-D part:*** For three-dimensional vectors, the dataset includes methods for calculating their magnitude, the cosine of the angle between two vectors, their dot product, and the horizontal coordinate of their cross product. In conjunction with point coordinates, it facilitates the calculation of the directed distance from a point to a plane, essential for spatial analysis.

In terms of solids, the dataset aids in finding the surface area and volume of cylinders using their base radius and height. The same parameters are used for cones, with additional calculations for their surface area and volume. For frustums, the dataset provides a method to determine the surface area and volume from the radii of the upper and lower bases and the height.

Spherical geometry is also covered, with the dataset enabling the calculation of a sphere's surface area from its radius. In the study of cuboids, the dataset allows for the determination of the sum of edge lengths, surface area, and volume from the lengths of the three edges. Similarly, for cubes, the side length can be used to find the sum of edge lengths, surface area, and volume.

The dataset also includes calculations for pyramids, using the base area and height to find the volume. For regular tetrahedrons, the base edge and height, or the base edge and side, provide the necessary measurements to calculate surface area and volume. In addition, the hardest ones show that the volume of an arbitrary tetrahedron can be calculated using two equations.

The complete set of symbolic equations can be found in Table 2, Appendix Section A.

Furthermore, the result from the determinant calculation may lead to misconceptions regarding the polynomial order within the models. A third-order determinant consists of three positive and three negative polynomials. The interplay between these positive and negative elements often misleads the model's search direction. Therefore, searching ability against bad equations are useful in this benchmark.

In section 3.3, we mentioned the growing difficulties of geometric equations, this comes the difficulty levels. And difficulty levels are based on baseline results, categorizing equations from simple polynomials to complex non-linear functions.

- **Easy:** This category contains the simplest equations, such as the perimeter of a triangle given the lengths of its three edges ($P = a + b + c$) and the volume of a pyramid given the base area and height ($V = \frac{1}{3}Sh$). In summary, this level comprises combinations of basic polynomial equations, making them relatively easy to solve. Each equation in this category can typically be solved within one hour.

- **Medium:** This category includes equations involving non-linear terms. Examples include the Pythagorean theorem ($c = \sqrt{a^2 + b^2}$) and finding the vertical coordinate of the intersection of two lines given their slopes and intercepts ($y = \frac{k_2 b_1 - k_1 b_2}{k_2 - k_1}$). While these equations introduce non-linear components, they remain closely related to basic polynomial structures. Solving each equation in this category typically requires approximately five hours.

- **Hard:** This category features the most complex equations, such as the volume of an arbitrary tetrahedron and Heron's formula for the area of a triangle based on the lengths of its three sides ($S = \sqrt{\frac{(a+b+c)(a+b-c)(a+c-b)(b+c-a)}{16}} = \sqrt{2a^2 b^2 + 2a^2 c^2 + 2b^2 c^2 - a^4 - b^4 - c^4}/4$). These equations are characterized by longer and deeper mathematical structures, making them significantly more challenging to solve. Each equation in this category typically requires up to one day to solve.

And our dataset contains two different sizes: 100000 for machine learning models that need to fit the curve and 500 for others.

## 3.2 METRICS

We use the symbolic recovery rate as the primary metric for evaluating performance, calculated as follows:

$$\text{recovery rate} = \frac{\text{count of successful discoveries}}{\text{count of total roll-outs}} \tag{1}$$

Running detail of this dataset is at section B. The metrics used for evaluating our benchmark are as follows:

- **Overall Recovery Rate:** The average recovery rate across all 71 datasets. This metric is designed to test the symbolic regression ability among all models.

- **Categorized Recovery Rate:** This metric allows for performance evaluation within specific difficulty levels (easy, medium, hard). By focusing on one category at a time, models can demonstrate their stability on easy problems and their capacity for exploration on hard problems.

- **Result-Oriented Recovery Rate:** Additional sub-categories can be created based on different dimensions, such as 2D versus 3D problems, the type of object studied (e.g., triangle, circle, sphere), and the type of result (e.g., perimeter, area, volume). This allows models to be compared within specific domains and contexts to highlight their performance in particular scenarios.

- **Number of Discovered Equations:** Since multiple runs can be performed for each algorithm, we also calculate the number of distinct expressions successfully discovered, where the recovery rate is greater than 0%. A higher number of discovered expressions reflects the model's ability to search effectively across different problem spaces.

### 3.3 MAJOR DIFFERENCE BETWEEN OUR DATASET AND SRBENCH

We think we have 5 major different from the SRbench:

- Purpose of the Dataset: Our dataset is designed to identify symbolic equations that are both simple and explainable to effectively solve problems. We have intentionally composed this dataset of ground truth equations rather than real-world scenarios that lack verifiable explanations for their functions. Consequently, traditional metrics like R-squared or other error measures are not applicable to our goals since they do not align with our focus on explainability.

- Patterns in Geometric: Unlike the Feynman dataset, which encompasses equations from diverse regions and subjects, our dataset is specifically focused on geometric data within a defined area. We concentrate on discovering patterns in geometric properties such as volume, area, and length. The primary motivation for selecting geometric equations is their inherent potential to unveil these patterns.

- Structured Learning Progression: The Feynman dataset includes a few sequences that progress from easy to difficult, such as the series from I.6.20 a to I.6.20 b. Our dataset, however, clearly illustrates many such progressions: for instance, from Helen's law to the calculation of circumcircle or incircle radii, which utilize Helen's law, or from Pythagoras' theorem to the cosine law, with the former being a special case of the latter. These process facilitates a deeper and more sequential learning experience.

- Realistic Constraints in Equations: Our dataset includes equations with generational constraints, such as the triangle constraint where the sum of two edges must exceed the third, and their difference must be less. These constraints make our data more realistic compared to data from SRbench, which is typically generated from uniform distributions. This approach ensures that our dataset not only supports the discovery of geometric relationships but also adheres more closely to real-world scenarios.

- Complex Equations with Few Inputs: Geometry excels at establishing intricate relationships between variables using a minimal number of tokens, as exemplified by Helen's law. In symbolic regression, inputs are often chosen or crafted through feature engineering to reduce their number, but this does not necessarily simplify the underlying relationships between them. Therefore, having complex equations with few inputs is crucial because it challenges the models to uncover deep relationships without relying on a large number of variables.

The results from our benchmark also differ from those of SRBench. Many baselines in SRBench focus primarily on the $R^2$ score, which may suggest they are better at fitting curves. However, their capability to accurately recover true symbolic equations is lacking. Moreover, thanks to the Structured Learning Progression, we are able to categorize these symbolic equations and assess model performance across different levels of difficulty. Additionally, the patterns in geometry enable us to evaluate each model's performance within specific patterns. This understanding allows us to select better baselines for future problem-solving involving these patterns.

### 3.4 SYMBOLIC REGRESSION METHODS

We use 20 different symbolic method based on 8 different approach. The correspondence is shown in the table below Table 1 and all parameter setting is at Table 5 in Appendix Section C:

- **Bayesian Machine Scientist** (Guimerà et al., 2020): This model determines the posterior probability of each expression from a corpus of mathematical expressions compiled from Wikipedia. The MCMC algorithm is then used to sample from the posterior distribution of expressions, generating new expressions based on these probabilities.

- **PSRN** (Ruan et al., 2024): A symbolic regression model that utilizes parallelized tree search (PTS) to discover mathematical expressions from data. PSRN employs GPU-accelerated parallel evaluation of symbolic expressions and implements efficient subtree reuse and caching. The model features a unique approach of selecting expressions based on minimum loss, followed by recursive symbolic backward derivation. Its core parallel symbolic regression module can integrate with various token generation methods.

Table 1: Correspondence between symbolic regression methods and approaches. BF stands for Brute Force Searching, DL denotes Deep Learning methods, DRL stands for Deep Reinforcement Learning methods, GP refers to Genetic Programming, Pretrain refers to methods using transformer modules for pretraining, Dimension refers to special methods targeting dimensional constraints and MCTS refers to machine learning models using the Monte Carlo Tree Search algorithm.

| Symbolic Regression Method | Category |
|---|---|
| Bayesian Machine Scientist | Bayesian |
| PSRN | BF |
| EQL | DL |
| AIFeynman | DL |
| NGGP | DRL, GP |
| uDSR | DRL, GP, Pretrain |
| PhySO | DRL, GP, Dimension |
| gplearn | GP |
| DEAP | GP |
| PySR | GP, Dimension |
| SINDy | Linear |
| SymINDy | Linear, GP |
| KAN | Linear |
| SPL | MCTS |
| RSRM | MCTS, GP |
| NeSymReS | Pretrain |
| E2E | Pretrain |
| DGSR | Pretrain, GP |
| TPSR | Pretrain, MCTS |
| SNIP | Pretrain |

- **EQL** (Martius & Lampert, 2016; Sahoo et al., 2018): This model uses multiplication units and nonlinear activation functions (e.g., sine and cosine) in its neural network. Each layer contains linear mappings and nonlinear transformations, and the network is trained using a Lasso-like objective function, combining L2 loss and L1 regularization.

- **AIFeynman** (Udrescu & Tegmark, 2020; Udrescu et al., 2020): This model employs a neural network to fit the data, then uses the network to identify relationships between variables, such as symmetry. After this, AiFeynman runs a brute-force search based on the extracted knowledge.

- **NGGP** (Mundhenk et al., 2021a): An upgraded version of DSR (Petersen et al., 2019), NGGP uses an RNN-based model through deep reinforcement learning to learn the distribution of expressions. It then fine-tunes these expressions using GP methods, focusing only on those that have been improved through fine-tuning.

- **uDSR** (Landajuela et al., 2022): An upgraded version of NGGP (Mundhenk et al., 2021a), this model incorporates the AiFeynman module to reduce the number of variables. It also introduces a linear token for generating polynomials and utilizes large-scale pretraining.

- **PhySO** (Tenachi et al., 2023): This model applies dimensional constraints to the NGGP (Mundhenk et al., 2021a) module. If a generated token violates dimensional constraints (e.g., summing variables with different dimensions), the generation probability is set to zero.

- **PySR** (Cranmer, 2023): Considered one of the best GP models, PySR optimizes hyperparameters algorithmically and supports dimensional constraints. When an expression violates dimensional constraints, its fitness is significantly penalized.

- **gplearn** (Stephens, 2016): This model retains the familiar scikit-learn fit/predict API, allowing it to work seamlessly with existing scikit-learn pipelines and grid search modules.

- **DEAP** (Fortin et al., 2012): A novel evolutionary computation framework designed for rapid prototyping and testing of ideas. It seeks to make algorithms explicit and data structures transparent. Many models using GP (Mundhenk et al., 2021a; Holt et al., 2022; Xu et al., 2024) rely on DEAP as their foundation.

- **SINDy** (Kaheman et al., 2020): The original SINDy model uses sparse regression techniques, such as LASSO, to obtain expressions from linear combinations of functions in a predefined library of candidate functions.

- **SymINDy** (Kitaitsev & Manzi, 2022): This model uses GP to generate libraries of candidate functions and integrates them with the SINDy method. The fitness value is positively correlated with the error produced by SINDy.

- **KAN** (Liu et al., 2024): In KAN, traditional weight parameters at the network's edges are replaced by univariate function parameters. Each node aggregates the outputs of these functions without any nonlinear transformations, relying on spline methods to replace traditional weight parameters.

- **SPL** (Sun et al., 2022): This model contains many predefined simple expressions as modules and uses the MCTS method to combine these modules into full expressions. After each roll-out, the best result is used as one of the modules for future iterations.

- **RSRM** (Xu et al., 2024): This model combines MCTS and GP to generate functions. It employs double Q-learning to initialize probabilities in the MCTS module, enabling the model to learn from previous roll-outs. The model also uses spline fitting to determine whether functions are odd or even and includes an MSDB block to extract useful modules from the best expressions for use in subsequent roll-outs.

- **NeSymReS** (Biggio et al., 2021): This model uses a pre-trained Transformer during the pre-training phase, trained on hundreds of millions of equations specifically generated for each batch. In the test step, an encoder encodes input expressions into latent vectors, from which the decoder iteratively samples candidate skeletons for the symbolic equation. For each candidate, numerical constants are fitted by treating them as independent parameters.

- **E2E** (Kamienny et al., 2022): This model trains a Transformer on a synthetic dataset to perform end-to-end (E2E) symbolic regression, directly predicting solutions without relying on skeletons. The predicted constants are refined using the BFGS algorithm (Roger Fletcher & Sons, 2013) as an informed starting point. Additionally, generative and inference techniques are introduced to allow the model to scale to larger problems.

- **DGSR** (Holt et al., 2022): This model trains a Transformer on a synthetic dataset, outputting expressions end-to-end, which are then refined using a GP module. The framework can perform symbolic regression on a large number of input variables while reducing computational cost during inference, as it encodes the data itself rather than the entire symbolic expression tree. This is achieved by learning representations of equations that capture invariant structures across different equations.

- **TPSR** (Shojaee et al., 2023): TPSR utilizes a forward planning algorithm that incorporates Monte Carlo Tree Search (MCTS) as a decoding strategy on top of a pre-trained Transformer-based SR model. This guides the generation of equation sequences. TPSR reduces overall inference time by incorporating feedback during the generation process and using an efficient caching mechanism.

- **SNIP** (Meidani et al., 2024): SNIP (Symbolic-Numeric Integrated Pre-training) bridges symbolic mathematical expressions and their corresponding numeric representations. The model employs dual Transformer encoders: one dedicated to learning symbolic representations and the other for numeric representations. Task-independent comparison targets enhance the similarity between the two representations. The multimodal pretraining of SNIP enables cross-modal understanding and generation of content.

## 4 RESULTS

We present two primary results derived from the measured datasets in Figure 3. Further details and additional results can be found in Appendix Section D Figure 4.

In the left panel, it is observed that the top five models in terms of recovery rate are RSRM (Xu et al., 2024), PSRN (Ruan et al., 2024), NGGP (Mundhenk et al., 2021a), PySR (Cranmer, 2023), and Bayesian Machine Scientist (Guimerà et al., 2020). Notably, methods based on deep learning and transformer-based pretraining tend to perform below these models.

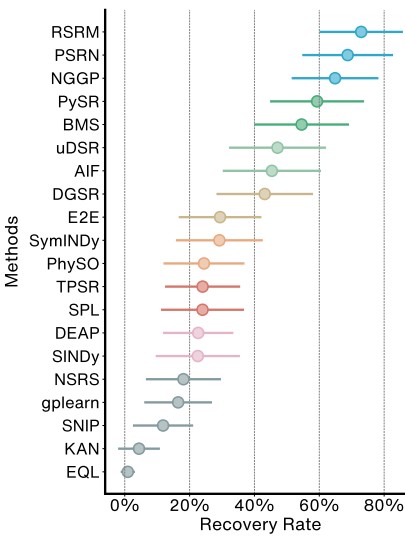 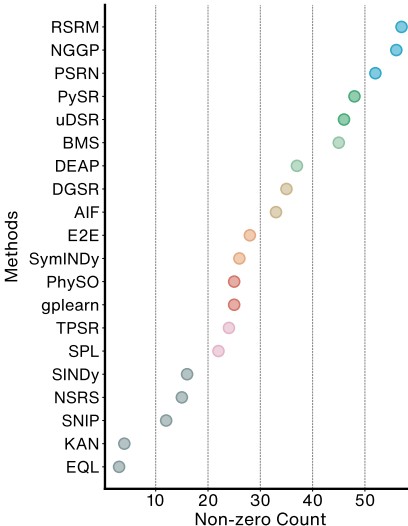

Figure 3: Results on the geometric dataset: the left panel illustrates the average recovery rate across all 71 equations, while the right panel displays the number of equations successfully discovered by the models. BMS, AIF and NGNS refer to the Bayesian Machine Scientist, AIFeynman and NeSymReS, respectively.

A comparison within the same methodological class reveals consistent improvements in performance over time. However, in the case of transformer-based pretraining methods, newer models such as SNIP (Meidani et al., 2024) demonstrate weaker performance compared to earlier models like End2End Transformers (Kamienny et al., 2022). This discrepancy could be attributed to a focus on optimizing the $R^2$ score, potentially at the expense of true symbolic regression capabilities.

In the genetic programming domain, PySR (Cranmer, 2023) significantly outperforms other models such as DEAP (Fortin et al., 2012) and gplearn (Stephens, 2016). While hyperparameter tuning may contribute to this performance difference, dimensional analysis also plays a crucial role. Specifically, PySR applies penalties to expressions that violate dimensional consistency, which improves the model's robustness. In contrast, PhySO (Tenachi et al., 2023) performs less effectively, ranking lower than both NGGP (Mundhenk et al., 2021a) and uDSR (Landajuela et al., 2022). PhySO's strict adherence to dimensional consistency dramatically reduces its search space, potentially leading to overfitting early in the training process.

The right panel of Figure 3 mirrors the trends observed in the left panel. While some models exhibit low recovery rates, they still manage to discover a significant number of equations, as exemplified by gplearn (Stephens, 2016).

## 5 CONCLUSION

In conclusion, we introduce a novel symbolic regression dataset, comprising a refined version of the SRBench dataset. We evaluate the performance of 20 different models across 8 methodological categories. Our analysis indicates that Monte Carlo Tree Search (MCTS) methods are particularly well-suited to this task, due to their broad search capabilities. Parallel search algorithms and deep reinforcement learning methods also demonstrate strong performance.

Furthermore, we highlight that an exclusive focus on optimizing the $R^2$ score can result in diminished symbolic recovery rates. As future work, we aim to identify additional symbolic equations for benchmarking and investigate optimal approaches for selecting equations under noisy conditions.

## ETHICS STATEMENT

Our studies does not involve human subjects, practices to data set releases, potentially harmful insights, methodologies and applications, pontential conflicts of interest and sponsorship, discrimination/bias/fairness concerns, privacy and security issues, legal compliance, and research integrity issues.

## REPRODUCIBILITY STATEMENT

Codes and models of Geometric Benchmark will be available at github upon the paper's publication. Details about experiments mentioned is at appendix section A, C.

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

## A  DATASET DETAILS

This section provides a detailed description of the geometric dataset. The complete dataset is presented in Tables 2, 3, and 4. The dataset comprises 8 parts: Dataset name, Equation, Category, Input data label, Input dimension, Output data label, Output dimension and Limitations. And our dataset contains two different sizes: 500 for normal model and 100000 for machine learning models that need to fit the curve.

- Dataset Name: This part specifies the name or identifier of the dataset, providing a clear reference for the specific set of geometric data being described according to the type of geometric shapes or phenomena it covers.

- Category: This section categorizes the dataset's difficulties. Easy polynomial expressions are classified to easy and complex polynomial with few non-linear tokens are classified to medium and other hard equations are classified to hard.

- Equation: This section lists the mathematical equations associated with the geometric shapes or phenomena covered in the dataset. These equations are used to compute various properties, such as volume, area, or perimeter, based on the input data.

- Input Data Label: This part describes the labels or names of the input variables. These labels indicate what each input represents, such as the length of an edge, the height, or the angles between edges in geometric shapes.

- Input Dimension: This section provides the dimensionality of the input data. It specifies the number of input variables or parameters required for the equations. For instance, a triangle might require two side lengths with $m$ dim and an angle between them with $rad$ dim.

- Output Data Label: This part describes the labels or names of the output variables. These labels indicate the properties being calculated, such as area, volume, or perimeter.

- Output Dimension: This section provides the dimensionality of the output data. It specifies the results generated by the equations. For instance, calculating the area of a rectangle results in output dimensions of $m^2$.

- Limitations: This part outlines any constraints or limitations associated with the dataset or the equations. These might include restrictions on the values of input parameters or specific conditions under which the equations are valid like the sum of two edges can not be larger than the other one in triangles.

The generation process follows this logic: values are randomly generated, with angles sampled uniformly from the interval $[0, \pi]$, and other values sampled uniformly from the range $[1, 5]$. The generated values are then evaluated against predefined constraints (Limitation from dataset). If any of these constraints are violated, new values are generated, and this process continues until the dataset size reaches either 500 or 100,000, depending on the specified target.

Table 2: 1 part/3 part of geometric dataset.

| Dataset name | category | Equation |
|---|---|---|
| triangle-1 | easy | $x_1 + x_2 + x_3$ |
| triangle-2 | hard | $\sqrt{-x_1^4 + 2x_1^2 x_2^2 + 2x_1^2 x_3^2 - x_2^4 + 2x_2^2 x_3^2 - x_3^4}/4$ |
| triangle-3 | hard | $x_1 + x_2 + \sqrt{x_1^2 - 2x_1 x_2 \cos(x_3) + x_2^2}$ |
| triangle-4 | medium | $(x_1 x_2 \sin(x_3))/2$ |
| triangle-5 | hard | $x_1 \sin(x_2)/\sin(x_2 + x_3) + x_1 \sin(x_3)/\sin(x_2 + x_3) + x_1$ |
| triangle-6 | hard | $x_1^2 \sin(x_2) \sin(x_3)/(2\sin(x_2 + x_3))$ |
| triangle-7 | medium | $x_1 + x_1 \sin(x_3)/\sin(x_2) + x_1 \sin(x_2 + x_3)/\sin(x_2)$ |
| triangle-8 | medium | $x_1^2 \sin(x_3) \sin(x_2 + x_3)/(2\sin(x_2))$ |
| triangle-9 | medium | $x_1 + x_2 + \sqrt{(-x_1^2 + x_2^2)}$ |
| triangle-10 | medium | $\sqrt{-x_1^4 + x_1^2 x_2^2}/2$ |
| triangle-11 | medium | $x_1 + x_1 \tan(x_2) + x_1/\cos(x_2)$ |
| triangle-12 | medium | $x_1^2 \tan(x_2)/2$ |
| triangle-13 | medium | $x_1 \sin(x_3)/\sin(x_2)$ |
| triangle-14 | hard | $\sqrt{x_1^2 - 2x_1 x_2 \cos(x_3) + x_2^2}$ |
| triangle-15 | medium | $\sqrt{x_1^2 + x_2^2}$ |
| triangle-16 | hard | $(x_1 x_2 x_3)/\sqrt{-x_1^4 + 2x_1^2 x_2^2 + 2x_1^2 x_3^2 - x_2^4 + 2x_2^2 x_3^2 - x_3^4}$ |
| triangle-17 | hard | $\sqrt{-x_1^4 + 2x_1^2 x_2^2 + 2x_1^2 x_3^2 - x_2^4 + 2x_2^2 x_3^2 - x_3^4}/(2x_1 + 2x_2 + 2x_3)$ |
| triangle-18 | easy | $(x_1 + x_3 + x_5)/2$ |
| triangle-19 | hard | $(x_1 \sqrt{x_3^2 + x_4^2} + x_3 \sqrt{(x_1 - x_3)^2 + (x_2 - x_4)^2})/(\sqrt{x_1^2 + x_2^2} + \sqrt{x_3^2 + x_4^2} + \sqrt{(x_1 - x_3)^2 + (x_2 - x_4)^2})$ |
| triangle-20 | hard | $(x_1^2 x_4 + x_2^2 x_4 - x_2 x_3^2 - x_2 x_4^2)/(2(x_1 x_4 - x_2 x_3))$ |
| triangle-21 | hard | $(-x_1 x_2 x_3 + x_1 x_3 x_4 - x_2^2 x_4 + x_2 x_4^2)/(x_1 x_4 - x_2 x_3)$ |
| circle-1 | easy | $2\pi x_1$ |
| circle-2 | easy | $\pi x_1^2$ |
| circle-3 | easy | $\pi x_1 x_2$ |
| circle-4 | medium | $\sqrt{x_1^2 - x_2^2}$ |
| circle-5 | easy | $(x_2 + 2)x_1$ |
| circle-6 | easy | $x_2 x_1^2/2$ |
| trapezoid-1 | hard | $x_1 + x_2 + \sqrt{x_1^2 - 2x_1 x_2 + x_2^2 + 4x_3^2}$ |
| trapezoid-2 | easy | $x_1 x_3/2 + x_2 x_3/2$ |
| trapezoid-3 | easy | $x_1 + x_2 + 2x_3$ |
| trapezoid-4 | hard | $(x_1 + x_2)\sqrt{-x_1^2 + 2x_1 x_2 - x_2^2 + 4x_3^2}/4$ |
| trapezoid-5 | medium | $x_1 - 2x_1/\cos(x_3) + x_2 + 2x_2/\cos(x_3)$ |
| trapezoid-6 | medium | $-x_1^2 \tan(x_3)/4 + x_2^2 \tan(x_3)/4$ |
| rectangle-1 | easy | $2x_1 + 2x_2$ |
| rectangle-2 | easy | $x_1 x_2$ |
| rectangle-3 | easy | $4x_1$ |
| rectangle-4 | easy | $x_1^2$ |
| rectangle-5 | easy | $2x_1 + 2x_2$ |
| rectangle-6 | medium | $x_1 x_2 \sin(x_3)$ |
| line-1 | medium | $(x_2 - x_4)/(x_3 - x_1)$ |
| line-2 | medium | $(x_3 x_2 - x_1 x_4)/(x_3 - x_1)$ |
| line-3 | medium | $(x_1 - x_3)/(1 - x_1 x_3)$ |
| line-4 | medium | $(x_3 - x_1)/(x_4 - x_2)$ |
| line-5 | medium | $(x_4 x_1 - x_2 x_3)/(x_1 - x_3)$ |
| line-6 | hard | $(x_2 - x_3 x_1 - x_4)/\sqrt{x_3^2 + 1}$ |
| vector3d-1 | medium | $\sqrt{x_1^2 + x_2^2 + x_3^2}$ |
| vector3d-2 | hard | $(x_1 x_2 + x_3 x_4 + x_5 x_6)/\sqrt{(x_1^2 + x_3^2 + x_5^2)(x_2^2 + x_4^2 + x_6^2)}$ |
| vector3d-3 | medium | $x_1 x_2 + x_3 x_4 + x_5 x_6$ |
| vector3d-4 | easy | $x_3 x_6 - x_4 x_5$ |
| vector3d-5 | hard | $(x_1 x_4 + x_2 x_5 + x_3 x_6)/\sqrt{x_4^2 + x_5^2 + x_6^2}$ |
| sphere-1 | easy | $\pi x_1^2 + 2\pi x_1 x_2$ |
| sphere-2 | easy | $\pi x_1^2 x_2$ |
| sphere-3 | hard | $\pi x_1^2 + 2\pi x_1 \sqrt{x_1^2 + x_2^2}$ |
| sphere-4 | easy | $\pi/3 x_1^2 x_2$ |
| sphere-5 | hard | $\pi(x_1^2 + x_2^2 + \sqrt{x_3^2 + (x_2 - x_1)^2})(x_1 + x_2)$ |
| sphere-6 | medium | $\pi(x_1^2 + x_1 x_2 + x_2^2)x_3$ |
| sphere-7 | easy | $4\pi/3 x_1^3$ |
| sphere-8 | easy | $4\pi x_1^2$ |
| cuboid-1 | easy | $4x_1 + 4x_2 + 4x_3$ |
| cuboid-2 | easy | $2x_1 x_2 + 2x_1 x_3 + 2x_2 x_3$ |
| cuboid-3 | easy | $x_1 x_2 x_3$ |
| cuboid-4 | easy | $12x_1$ |
| cuboid-5 | easy | $6x_1^2$ |
| cuboid-6 | easy | $x_1^3$ |
| regular-tetrahedron-1 | medium | $x_1 \sqrt{4x_2^2 + x_1^2} + x_1^2$ |
| regular-tetrahedron-2 | medium | $1/3 x_1^2 \sqrt{x_2^2 - \frac{1}{2}x_1^2}$ |
| regular-tetrahedron-3 | easy | $1/3 x_1^2 x_2$ |
| tetrahedron-1 | hard | $1/12 x_4 \sqrt{-x_1^4 + 2x_1^2 x_2^2 + 2x_1^2 x_3^2 - x_2^4 + 2x_2^2 x_3^2 - x_3^4}$ |
| tetrahedron-2 | easy | $x_1 x_2/3$ |
| tetrahedron-3 | hard | $1/6 x_1 x_2 x_3 \sqrt{\sin(x_4)^2 + \sin(x_5)^2 + \sin(x_6)^2 + 2\cos(x_4)\cos(x_5)\cos(x_6) - 2}$ |
| tetrahedron-4 | hard | $1/3 \sqrt{(-x_1^2/2 + x_2^2/2 + x_3^2/2)(x_1^2/2 - x_2^2/2 + x_3^2/2)(x_1^2/2 + x_2^2/2 - x_3^2/2)}$ |

Table 3: 2 part/3 part of geometric dataset.

| Dataset name | Input data label | Input dimensions |
|---|---|---|
| triangle-1 | Triangle three sides | m m m |
| triangle-2 | Triangle three sides | m m m |
| triangle-3 | Triangle two sides and the included angle | m m r |
| triangle-4 | Triangle two sides and the included angle | m m r |
| triangle-5 | Triangle two angles and the included side | m r r |
| triangle-6 | Triangle two angles and the included side | m r r |
| triangle-7 | Triangle two angles and the opposite sides | m r r |
| triangle-8 | Triangle two angles and the opposite sides | m r r |
| triangle-9 | Right-angled triangle right sides and hypotenuse | m m |
| triangle-10 | Right-angled triangle right sides and hypotenuse | m m |
| triangle-11 | Right-angled triangle right side and opposite angle | m r |
| triangle-12 | Right-angled triangle right side and opposite angle | m r |
| triangle-13 | Triangle two angles and the opposite sides | m r r |
| triangle-14 | Triangle two sides and the included angle | m m r |
| triangle-15 | Right-angled triangle two right sides | m m |
| triangle-16 | Triangle three sides | m m m |
| triangle-17 | Triangle three sides | m m m |
| triangle-18 | Triangle three points' coordinates | m m m m m m |
| triangle-19 | Triangle two points' coordinates | m m m m |
| triangle-20 | Triangle two points' coordinates | m m m m |
| triangle-21 | Triangle two points' coordinates | m m m m |
| circle-1 | Circle radius | m |
| circle-2 | Circle radius | m |
| circle-3 | Ellipse major and minor axis | m m |
| circle-4 | Ellipse major and minor axis | m m |
| circle-5 | Sector radius and angle | m r |
| circle-6 | Sector radius and angle | m r |
| trapezoid-1 | Isosceles trapezoid upper base lower base and height | m m m |
| trapezoid-2 | Isosceles trapezoid upper base lower base and height | m m m |
| trapezoid-3 | Isosceles trapezoid upper base lower base and side | m m m |
| trapezoid-4 | Isosceles trapezoid upper base lower base and side | m m m |
| trapezoid-5 | Isosceles trapezoid upper base lower base and side angle | m m r |
| trapezoid-6 | Isosceles trapezoid upper base lower base and side angle | m m r |
| rectangle-1 | Rectangle two sides | m m |
| rectangle-2 | Rectangle two sides | m m |
| rectangle-3 | Square side length | m |
| rectangle-4 | Square side length | m |
| rectangle-5 | Parallelogram two sides and included angle | m m r |
| rectangle-6 | Parallelogram two sides and included angle | m m r |
| line-1 | Two lines slope and intercept | 1 m 1 m |
| line-2 | Two lines slope and intercept | 1 m 1 m |
| line-3 | Two lines slope and intercept | 1 m 1 m |
| line-4 | Two points horizontal and vertical coordinates | m m m m |
| line-5 | Two points horizontal and vertical coordinates | m m m m |
| line-6 | Point horizontal and vertical coordinate and Line slope and intercept | m m 1 m |
| vector3d-1 | Three-dimensional vector | m m m |
| vector3d-2 | Two three-dimensional vectors | m m m m m m |
| vector3d-3 | Two three-dimensional vectors | m m m m m m |
| vector3d-4 | Two three-dimensional vectors | m m m m m m |
| vector3d-5 | Three-dimensional vector and point coordinates | m m m 1 1 1 |
| sphere-1 | Cylinder base radius and height | m m |
| sphere-2 | Cylinder base radius and height | m m |
| sphere-3 | Cone base radius and height | m m |
| sphere-4 | Cone base radius and height | m m |
| sphere-5 | Frustum upper and lower base radius and height | m m m |
| sphere-6 | Frustum upper and lower base radius and height | m m m |
| sphere-7 | Sphere radius | m |
| sphere-8 | Sphere radius | m |
| cuboid-1 | Cuboid three edge lengths | m m m |
| cuboid-2 | Cuboid three edge lengths | m m m |
| cuboid-3 | Cuboid three edge lengths | m m m |
| cuboid-4 | Cube side length | m |
| cuboid-5 | Cube side length | m |
| cuboid-6 | Cube side length | m |
| regular tetrahedron-1 | Regular tetrahedron base edge and height | m m |
| regular tetrahedron-2 | Regular tetrahedron base edge and side | m m |
| regular tetrahedron-3 | Regular tetrahedron base edge and height | m m |
| tetrahedron-1 | tetrahedron three edges and height | m m m m |
| tetrahedron-2 | tetrahedron base area and height | $m^2$ m |
| tetrahedron-3 | tetrahedron three edges and three angles from one point | m m m r r r |
| tetrahedron-4 | isohedral tetrahedron 3 edges | m m m |

Table 4: 3 part/3 part of geometric dataset.

| Dataset name | Output data label | Output dimension | Limitations |
|---|---|---|---|
| triangle-1 | Perimeter | m | $x_1 + x_2 > x_3, x_1 + x_3 > x_2, x_2 + x_3 > x_1$ |
| triangle-2 | Area | m$^2$ | $x_1 + x_2 > x_3, x_1 + x_3 > x_2, x_2 + x_3 > x_1$ |
| triangle-3 | Perimeter | m | |
| triangle-4 | Area | m$^2$ | |
| triangle-5 | Perimeter | m | $x_2 + x_3 < \pi$ |
| triangle-6 | Area | m$^2$ | $x_2 + x_3 < \pi$ |
| triangle-7 | Perimeter | m | $x_2 + x_3 < \pi$ |
| triangle-8 | Area | m$^2$ | $x_2 + x_3 < \pi$ |
| triangle-9 | Perimeter | m | $x_1 < x_2$ |
| triangle-10 | Area | m$^2$ | $x_1 < x_2$ |
| triangle-11 | Perimeter | m | $x_2 < \pi/2$ |
| triangle-12 | Area | m$^2$ | $x_2 < \pi/2$ |
| triangle-13 | Another side | m | $x_2 + x_3 < \pi$ |
| triangle-14 | Another side | m | |
| triangle-15 | Hypotenuse | m | |
| triangle-16 | Circumcircle radius | m | $x_1 + x_2 > x_3, x_1 + x_3 > x_2, x_2 + x_3 > x_1$ |
| triangle-17 | Incircle radius | m | $x_1 + x_2 > x_3, x_1 + x_3 > x_2, x_2 + x_3 > x_1$ |
| triangle-18 | Centroid horizontal coordinate | m | |
| triangle-19 | incenter horizontal coordinate | m | |
| triangle-20 | circumcenter horizontal coordinate | m | |
| triangle-21 | orthocenter horizontal coordinate | m | |
| circle-1 | Perimeter | m | |
| circle-2 | Area | m$^2$ | |
| circle-3 | Area | m$^2$ | $x_1 > x_2$ |
| circle-4 | Focal point | m | $x_1 > x_2$ |
| circle-5 | Perimeter | m | |
| circle-6 | Area | m | |
| trapezoid-1 | Perimeter | m | |
| trapezoid-2 | Area | m$^2$ | |
| trapezoid-3 | Perimeter | m | $x_3 > (x_1 - x_2)/2, x_3 > (x_2 - x_1)/2$ |
| trapezoid-4 | Area | m$^2$ | $x_3 > (x_1 - x_2)/2, x_3 > (x_2 - x_1)/2$ |
| trapezoid-5 | Perimeter | m | $x_1 < x_2, x_3 < \pi/2$ |
| trapezoid-6 | Area | m$^2$ | $x_1 < x_2, x_3 < \pi/2$ |
| rectangle-1 | Perimeter | m | |
| rectangle-2 | Area | m$^2$ | |
| rectangle-3 | Perimeter | m | |
| rectangle-4 | Area | m$^2$ | |
| rectangle-5 | Perimeter | m | |
| rectangle-6 | Area | m$^2$ | |
| line-1 | Intersection horizontal coordinate | m | |
| line-2 | Intersection vertical coordinate | m | |
| line-3 | Angle tangent value | 1 | |
| line-4 | Slope of the line through two points | 1 | |
| line-5 | Intercept of the line through two points | m | |
| line-6 | Point to line distance (directed) | m | |
| vector3d-1 | Magnitude | m | |
| vector3d-2 | Cosine value of the angle | m | |
| vector3d-3 | Dot product | m | |
| vector3d-4 | Cross product horizontal coordinate | m | |
| vector3d-5 | Point to plane distance (directed) | m | |
| sphere-1 | Surface Area | m$^2$ | |
| sphere-2 | Volume | m$^3$ | |
| sphere-3 | Surface Area | m$^2$ | |
| sphere-4 | Volume | m$^3$ | |
| sphere-5 | Surface Area | m$^2$ | |
| sphere-6 | Volume | m$^3$ | |
| sphere-7 | Surface Area | m$^3$ | |
| sphere-8 | Surface Area | m$^2$ | |
| cuboid-1 | Sum of edge lengths | m | |
| cuboid-2 | Surface Area | m$^2$ | |
| cuboid-3 | Volume | m$^3$ | |
| cuboid-4 | Sum of edge lengths | m | |
| cuboid-5 | Surface Area | m$^2$ | |
| cuboid-6 | Volume | m$^3$ | |
| regular tetrahedron-1 | Surface Area | m$^2$ | |
| regular tetrahedron-2 | Volume | m$^3$ | $x_2^2 > \frac{1}{2}x_1^2$ |
| regular tetrahedron-3 | Volume | m$^3$ | |
| tetrahedron-1 | Volume | m$^3$ | $x_1 + x_2 > x_3, x_1 + x_3 > x_2, x_2 + x_3 > x_1$ |
| tetrahedron-2 | Volume | m$^3$ | |
| tetrahedron-3 | Volume | m$^3$ | $x_4 + x_5 + x_6 < \pi$ |
| tetrahedron-4 | Volume | m$^3$ | $x_1 + x_2 > x_3, x_1 + x_3 > x_2, x_2 + x_3 > x_1$ |

# B  SYMBOLIC EQUIVALENT ALGORITHMS

The method for distinguishing a successful discovery is outlined in Algorithm 1. We choose sympy (Meurer et al., 2017) to simplify the expression and human justify. We conduct 100 independent runs with different random seeds, and the time limits for the easy, medium, and hard problems are set to 1 hour, 5 hours, and 24 hours, respectively. Additionally, the hardware constraints include 10 CPU cores and one A100 GPU.

We create a new algorithm to fix the wrong judgment of symbolic equations in SRbench (La Cava et al., 2021), since they consider $m_0 * v/sqrt(1 - v**2/c**2)$ and $m_0 ** 1.5 * v/(m_0 * (-v ** 2/c ** 2 + 1.0)) ** 0.5$ are different equations and they might ignore equations symbolic error more than $10^{-3}$.

---

**Algorithm 1** Algorithm for Discriminating the Correct Expression

---

**Input:** dataset $\mathcal{S}_{data} = (X, y)$, ground truth expression $\mathcal{F}$, input expression $\mathcal{F}_i$, simplify function.
**Output:** Boolean value representing whether the input expression is correct.
    $\mathcal{F}_i(X) \to \hat{y}$                  ▷ Evaluate the input expression $\mathcal{F}_i$ on $X$ to obtain $\hat{y}$
    $||y - \hat{y}|| \to err$              ▷ Compute error between predicted and actual values
    **if** $err \geq 10^{-5}$ **then**
        **return** false                  ▷ Return false if error exceeds threshold
    **end if**
    $simplify(\mathcal{F}_i) \to \mathcal{F}_i$                ▷ Simplify the input expression
    $\mathcal{F}_i - \mathcal{F} \to \mathcal{G}$    ▷ Compute the difference functions between input and ground truth expressions
    $simplify(\mathcal{G}) \to \mathcal{G}$            ▷ remove redundant sub-expressions
    replace constants below $10^{-5}$ in $\mathcal{G}$ with 0
    **if** $\mathcal{G}$ is empty **then**
        **return** true               ▷ Return true if the expressions are equivalent
    **end if**
    $\mathcal{G}(X) \to \hat{z}$                     ▷ Evaluate $\mathcal{G}$ on $X$ to obtain $\hat{z}$
    $||\hat{z}|| \to err$                ▷ Compute error for the difference expression
    **if** $err \geq 10^{-20}$ **then**
        **return** human_justify($\mathcal{G}$)     ▷ If error is still significant, defer to human justification
    **end if**
    **return** true                  ▷ Return true if the difference is negligible

---

## C  MODEL DETAILS

In this section, we give hyper-parameters of all 20 models at Table 5. The other parameters not mentioned in table is set as default value. In most models (transformer models might be better with their pre-training stage tokens.), the token set is $+, -, \times, \div$ and $\cos, \sin, \sqrt{\cdot}$ and $X, const$.

Table 5: Hyper-parameter setting of all 20 models.

| Model | Hyper-parameters |
|---|---|
| Bayesian Machine Scientist | { Drtarget: 60, nsample: 1000, anneal: 20, burnin: 5000, annealf: 6 } |
| PSRN | { trying_const_num: 2,trying_const_range: [0,4], trying_const_n_try:3 } |
| EQL | { $l_0$_reg: 0.0001, iterations: 10 } |
| AIFeynman | { BF_try_time: 60, BF_ops_file_type: "14ops", polyfit_deg: 3, NN_epochs: 1000 } |
| NGGP | { gp_population_size: 500, generations: 20, p_crossover: 0.5, p_mutate: 0.5, tournament_size: 5, train_n: 50, mutate_tree_max: 3, n_samples: 200000, batch_size: 500 } |
| uDSR | { function_set: [add, sub, mul, div, sin, cos, sqrt, const, poly], poly_degree: 3, gp_population_size: 500, generations: 20, p_crossover: 0.5, p_mutate: 0.5, tournament_size: 5, train_n: 50, mutate_tree_max: 3, n_samples: 200000, batch_size: 500 } |
| PhySO | { fixed_consts: [1, pi], fixed_consts_units: [[0], [0]], free_consts_names: [], free_consts_units : [], op_names: [mul, add, sub, div, inv, n2, sqrt, neg, sin, cos], run_config: config2.config2 } |
| gplearn | { population_size: 1000, generations: 20, p_crossover: 0.7 ,max_samples: 0.9, parsimony_coefficient: 0.01 } |
| DEAP | { const_range: (0,4), generations: 400, p_crossover: 0.3, p_mutate: 300} |
| PySR | { niterations: 200, weight_optimize: 0.001, adaptive_parsimony_scaling: 1000, parsimony: 0.0 } |
| SINDy | { library: GeneralizedLibrary([PolynomialLibrary, FourierLibrary]) degree: [2,3,4,5]} |
| SymINDy | { sparsity_coef: 0.01, library_name: "generalized", ngen: 20 } |
| KAN | { width: [num_of_inputs,2,1], grid: 3, k: 3 } |
| SPL | { transplant_step: 10000 } |
| RSRM | { tournsize: 10, max_height: 10, max_const: 6, cxpb: 0.1, mutpb: 0.5, pops: 500, times: 30, hof_size: 20, token_discount: 0.99, max_expr_num: 20, expr_ratio: 0.1, token_ratio": 0.5, form_type: [Add] } |
| NeSymReS | { config_file: "100M/eq_setting.json" } |
| E2E | { beam_size: 10, n_trees_to_refine: 10 max_input_points: 200, eval_input_length_modulo 50, prediction_sigmas: 1,2,4,8,16 } |
| DGSR | { training_equations: 200000, training_epochs: 20, batch_outer_datasets: 24, batch_inner_equations: 100, other_setting_file: "config.yaml" } |
| TPSR | { lam: 0.1, horizon: 200 width: 5 num_beams: 2, rollout: 5 max_input_points :200, max_number_bags :10 } |
| SNIP | { max_input_points: 200, lso_optimizer: gwo, lso_pop_size: 50, lso_max_iteration: 10, lso_stop_r2: 0.999, beam_size: 2 } |

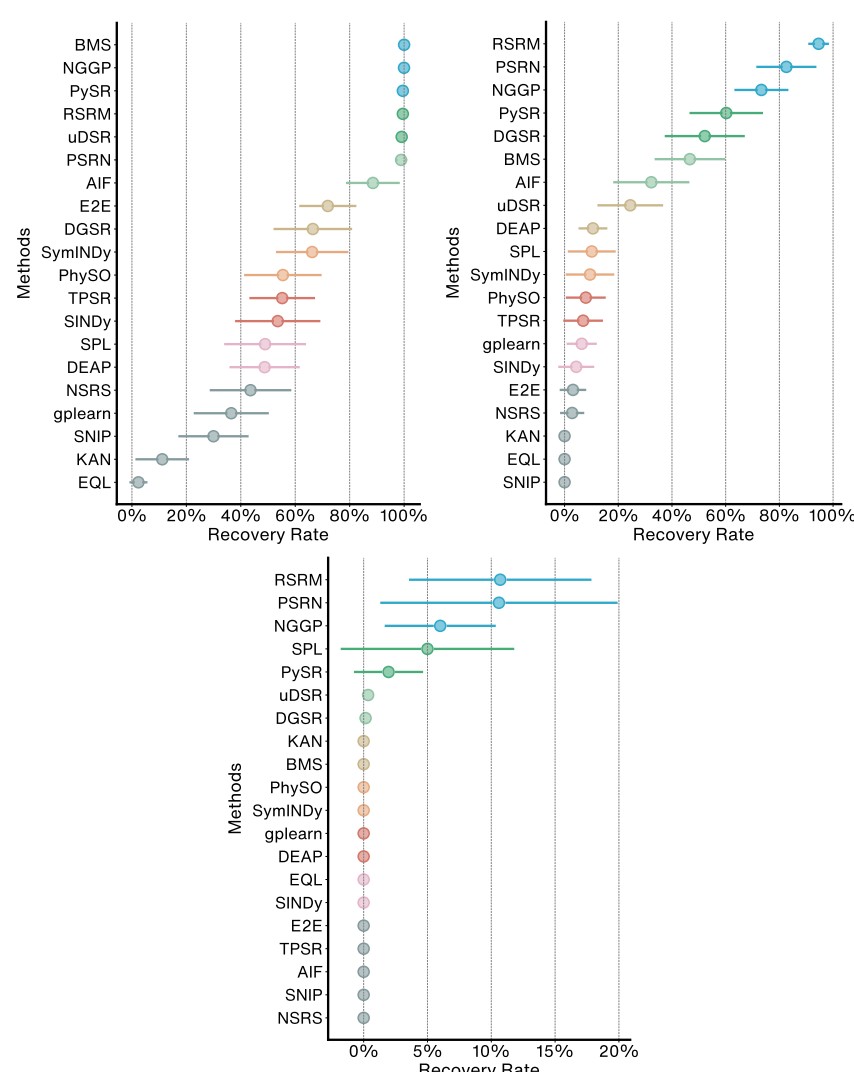

Figure 4: extra result within three different category.

## D    EXTRA RESULT

In this section, we present additional results from the geometric dataset. As shown in Figure 4, the performance of many models across the three difficulty levels—easy, medium, and hard—appears consistent. The strong symbolic regression capabilities demonstrated by models such as RSRM and PSRN can be attributed to their proficiency in handling medium and hard-level expressions. RSRM utilizes MSDB, a mechanism for storing previously encountered failure cases, while PSRN systematically explores a vast array of potential equations. This figure illustrates that both strategies are effective in improving symbolic regression performance.

Additionally, Bayesian models, such as the Bayesian Machine Scientist, achieve a 100% success rate in the easy category, highlighting their stability and reliability in simpler tasks.

## E    FULL RECOVERY SCORE OF EACH MODEL

In this section, we provide the recovery rate within each method and each model in Table 6.

Table 6: The recovery rate within each method and each model. BMS, AIF and NGNS refer to the Bayesian Machine Scientist, AIFeynman and NeSymReS, respectively.

| Dataset name | PSRN | PySR | NGGP | uDSR | RSRM | KAN | BMS | phySO | SymINDy | gplearn | DEAP | EQL | SINDy | SPL | E2E | TPSR | AIF | SNIP | DGSR | NSRS |
|---|---|---|---|---|---|---|---|---|---|---|---|---|---|---|---|---|---|---|---|---|
| triangle-1 | **1** | **1** | **1** | **1** | **1** | **1** | **1** | **1** | **1** | **1** | 0.95 | 0.5 | **1** | **1** | **1** | **1** | **1** | **1** | **1** | **1** |
| triangle-2 | 0 | 0 | 0 | 0 | 0 | 0 | 0 | 0 | 0 | 0 | 0 | 0 | 0 | 0 | 0 | 0 | 0 | 0 | 0 | 0 |
| triangle-3 | 0 | 0 | 0 | 0 | **0.12** | 0 | 0 | 0 | 0 | 0 | 0 | 0 | 0 | 0 | 0 | 0 | 0 | 0 | 0 | 0 |
| triangle-4 | **1** | **1** | **1** | 0.2 | **1** | 0 | 0.95 | 0 | **1** | 0.1 | 0.21 | 0 | 0 | 0.43 | 0 | 0 | **1** | 0 | 0.93 | 0 |
| triangle-5 | 0.12 | 0 | 0 | 0 | **0.19** | 0 | 0 | 0 | 0 | 0 | 0 | 0 | 0 | 0 | 0 | 0 | 0 | 0 | 0.03 | 0 |
| triangle-6 | **1** | 0 | 0.07 | 0 | 0.17 | 0 | 0 | 0 | 0 | 0 | 0 | 0 | 0 | 0 | 0 | 0 | 0 | 0 | 0 | 0 |
| triangle-7 | 0.07 | 0.78 | 0.21 | 0 | **1** | 0 | 0 | 0 | 0 | 0 | 0 | 0 | 0 | 0 | 0 | 0 | 0 | 0 | 0.65 | 0 |
| triangle-8 | **1** | 0.12 | 0.18 | 0 | **1** | 0 | 0 | 0 | 0 | 0 | 0.02 | 0 | 0 | 0 | 0 | 0 | 0 | 0 | 0.03 | 0 |
| triangle-9 | 0.67 | 0.27 | **1** | 0.04 | 0.71 | 0 | 0.05 | 0 | 0 | 0 | 0 | 0 | 0 | 0 | 0 | 0 | 0 | 0 | 0 | 0 |
| triangle-10 | **1** | **1** | **1** | 0.22 | **1** | 0 | 0.56 | 0 | 0 | 0 | 0 | 0 | 0 | 0 | 0 | 0 | 0.73 | 0 | 0 | 0 |
| triangle-11 | **1** | 0.93 | 0.99 | 0.01 | 0.66 | 0 | 0.71 | 0 | 0 | 0.01 | 0 | 0 | 0 | 0 | 0 | 0 | 0 | 0 | **1** | 0 |
| triangle-12 | **1** | 0.12 | 0.95 | 0 | **1** | 0 | **1** | 0 | 0 | 0.01 | 0.02 | 0 | 0 | 0 | 0 | 0 | 0 | 0 | **1** | 0 |
| triangle-13 | **1** | 0.78 | 0.97 | 0.02 | **1** | 0 | 0.75 | 0.06 | 0.1 | 0.2 | 0.29 | 0 | 0 | 0 | 0 | 0 | 0 | 0 | **1** | 0 |
| triangle-14 | 0 | 0 | **0.01** | 0 | 0 | 0 | 0 | 0 | 0 | 0 | 0 | 0 | 0 | 0 | 0 | 0 | 0 | 0 | 0 | 0 |
| triangle-15 | **1** | 0.83 | **1** | **1** | **1** | 0 | 0.95 | 0.71 | 0 | 0.07 | 0 | 0 | 0 | **1** | 0 | 0 | **1** | 0 | 0 | 0 |
| triangle-16 | 0 | 0 | 0 | 0 | 0 | 0 | 0 | 0 | 0 | 0 | 0 | 0 | 0 | 0 | 0 | 0 | 0 | 0 | 0 | 0 |
| triangle-17 | 0 | 0 | 0 | 0 | 0 | 0 | 0 | 0 | 0 | 0 | 0 | 0 | 0 | 0 | 0 | 0 | 0 | 0 | 0 | 0 |
| triangle-18 | 0.69 | 0.98 | **1** | **1** | **1** | **1** | **1** | 0 | 0.45 | 0 | 0 | 0.05 | **1** | 0 | 0.21 | 0 | 0 | 0 | 0.03 | 0 |
| triangle-19 | 0 | 0 | 0 | 0 | 0 | 0 | 0 | 0 | 0 | 0 | 0 | 0 | 0 | 0 | 0 | 0 | 0 | 0 | 0 | 0 |
| triangle-20 | 0 | 0 | 0 | 0 | 0 | 0 | 0 | 0 | 0 | 0 | 0 | 0 | 0 | 0 | 0 | 0 | 0 | 0 | 0 | 0 |
| triangle-21 | 0 | 0 | 0 | 0 | 0 | 0 | 0 | 0 | 0 | 0 | 0 | 0 | 0 | 0 | 0 | 0 | 0 | 0 | 0 | 0 |
| circle-1 | **1** | **1** | **1** | **1** | **1** | 0 | **1** | 0.93 | 0 | 0 | 0.15 | 0 | 0 | **1** | 0.77 | **1** | **1** | **1** | 0 | **1** |
| circle-2 | **1** | **1** | **1** | **1** | **1** | 0 | **1** | **1** | 0 | 0 | 0.06 | 0 | 0 | **1** | 0.95 | 0.73 | **1** | **1** | 0 | 0.34 |
| circle-3 | **1** | **1** | **1** | **1** | **1** | 0 | **1** | **1** | 0 | 0 | 0.02 | 0 | 0 | **1** | **1** | 0.87 | **1** | 0 | 0 | 0.12 |
| circle-4 | **1** | **1** | **1** | **1** | 0.76 | 0 | 0.71 | 0.15 | 0 | 0.13 | 0.07 | 0 | 0 | 0.9 | 0 | **1** | **1** | 0 | 0 | 0 |
| circle-5 | **1** | **1** | **1** | **1** | **1** | 0 | **1** | 0.92 | **1** | 0.89 | 0.85 | 0 | **1** | 0.1 | 0.99 | 0.53 | **1** | 0 | **1** | 0 |
| circle-6 | **1** | **1** | **1** | 0.94 | **1** | 0 | **1** | 0.17 | **1** | 0.02 | 0.57 | 0 | 0 | 0 | 0.17 | 0.53 | **1** | 0 | **1** | 0 |
| trapezoid-1 | 0 | 0 | 0.26 | 0.03 | **0.79** | 0 | 0 | 0 | 0 | 0 | 0 | 0 | 0 | 0 | 0 | 0 | 0 | 0 | 0 | 0 |
| trapezoid-2 | **1** | **1** | **1** | **1** | **1** | 0 | **1** | 0 | **1** | 0 | 0.13 | 0 | **1** | 0 | **1** | 0.27 | **1** | 0 | 0.73 | 0 |
| trapezoid-3 | **1** | **1** | **1** | **1** | **1** | 0 | **1** | **1** | **1** | 0.96 | 0.87 | 0 | **1** | **1** | 0.82 | 0.67 | **1** | 0.17 | **1** | 0 |
| trapezoid-4 | 0 | 0 | 0 | 0 | 0 | 0 | 0 | 0 | 0 | 0 | 0 | 0 | 0 | 0 | 0 | 0 | 0 | 0 | 0 | 0 |
| trapezoid-5 | **1** | 0 | 0.31 | 0 | **1** | 0 | 0 | 0 | 0 | 0.09 | 0 | 0 | 0 | 0 | 0 | 0 | 0 | 0 | 0.45 | 0 |
| trapezoid-6 | **1** | 0 | 0.24 | 0 | **1** | 0 | 0.22 | 0 | 0 | 0 | 0 | 0 | 0 | 0 | 0 | 0 | 0 | 0 | 0 | 0 |
| rectangle-1 | **1** | **1** | **1** | **1** | **1** | **1** | **1** | 0.9 | **1** | 0.95 | 0.94 | 0.12 | **1** | **1** | 0.61 | 0.4 | **1** | 0.17 | **1** | **1** |
| rectangle-2 | **1** | **1** | **1** | **1** | **1** | 0 | **1** | **1** | **1** | 0.98 | 0.96 | 0 | **1** | **1** | 0.99 | 0.87 | **1** | 0 | **1** | **1** |
| rectangle-3 | **1** | **1** | **1** | **1** | **1** | 0 | **1** | 0.8 | **1** | 0.76 | 0.98 | 0 | **1** | **1** | 0.86 | 0.8 | **1** | 0.75 | **1** | **1** |
| rectangle-4 | **1** | **1** | **1** | **1** | **1** | 0 | **1** | **1** | **1** | 0.98 | **1** | 0 | **1** | **1** | 0.71 | **1** | **1** | 0.4 | **1** | **1** |
| rectangle-5 | **1** | **1** | **1** | **1** | **1** | 0 | **1** | **1** | **1** | 0.93 | 0.96 | 0 | **1** | **1** | 0.12 | 0 | **1** | 0.5 | **1** | 0.74 |
| rectangle-6 | **1** | **1** | **1** | 0.26 | **1** | 0 | **1** | 0.9 | **1** | 0.82 | 0.72 | 0 | 0 | 0 | 0 | 0 | **1** | 0 | **1** | 0.65 |
| line-1 | **1** | 0.98 | 0.76 | 0 | **1** | 0 | 0.15 | 0 | 0 | 0.02 | 0.09 | 0 | 0 | 0 | 0 | 0 | 0 | 0 | **1** | 0 |
| line-2 | **1** | **1** | 0.56 | 0 | **1** | 0 | 0.15 | 0 | 0 | 0 | 0.24 | 0 | 0 | 0 | 0 | 0 | 0 | 0 | **1** | 0 |
| line-3 | **1** | 0.95 | **1** | 0.01 | **1** | 0 | 0.33 | 0 | 0 | 0 | 0.2 | 0 | 0 | 0 | 0 | 0 | 0 | 0 | 0.95 | 0 |
| line-4 | **1** | **1** | 0.93 | 0.01 | **1** | 0 | 0 | 0 | 0 | 0.03 | 0.19 | 0 | 0 | 0 | 0 | 0 | 0 | 0 | **1** | 0 |
| line-5 | **1** | 0.97 | 0.44 | 0.02 | **1** | 0 | 0 | 0 | 0 | 0 | 0.08 | 0 | 0 | 0 | 0 | 0 | 0 | 0 | **1** | 0 |
| line-6 | **1** | 0.39 | 0.51 | 0.04 | 0.12 | 0 | 0 | 0 | 0 | 0 | 0 | 0 | 0 | 0 | 0 | 0 | 0 | 0 | 0 | 0 |
| vector3d-1 | 0.26 | 0.03 | 0.32 | **1** | 0.93 | 0 | 0.2 | 0 | 0 | 0 | 0 | 0 | 0 | 0 | 0 | 0 | **1** | 0 | 0 | 0 |
| vector3d-2 | 0 | 0 | 0 | 0 | 0 | 0 | 0 | 0 | 0 | 0 | 0 | 0 | 0 | 0 | 0 | 0 | 0 | 0 | 0 | 0 |
| vector3d-3 | **1** | 0.12 | 0.42 | **1** | **1** | 0 | **1** | 0 | 0.08 | 0.08 | 0.21 | 0 | **1** | 0 | 0.72 | 0.92 | **1** | 0 | **1** | 0 |
| vector3d-4 | **1** | 0.98 | **1** | **1** | **1** | 0 | **1** | 0 | 0.23 | 0.15 | 0.36 | 0 | **1** | 0 | 0.01 | 0 | 0 | 0 | **1** | 0 |
| vector3d-5 | 0 | 0 | 0 | 0 | 0 | 0 | 0 | 0 | 0 | 0 | 0 | 0 | 0 | 0 | 0 | 0 | 0 | 0 | 0 | 0 |
| sphere-1 | **1** | **1** | **1** | **1** | **1** | 0 | **1** | 0.29 | 0.75 | 0 | 0 | 0 | 0 | 0 | 0.9 | 0.6 | **1** | 0 | 0 | 0 |
| sphere-2 | **1** | **1** | **1** | 0.95 | **1** | 0 | **1** | **1** | 0.43 | 0 | 0.04 | 0 | 0 | 0.2 | 0.68 | **1** | **1** | 0 | 0 | 0 |
| sphere-3 | 0 | 0 | 0.35 | 0 | **0.75** | 0 | 0 | 0 | 0 | 0 | 0 | 0 | 0 | 0 | 0 | 0 | 0 | 0 | 0 | 0 |
| sphere-4 | **1** | **1** | **1** | 0.99 | **1** | 0 | **1** | 0 | 0 | 0 | 0 | 0 | 0 | 0 | 0.59 | 0.6 | **1** | 0 | 0 | 0 |
| sphere-5 | 0 | 0 | 0 | 0 | 0 | 0 | 0 | 0 | 0 | 0 | 0 | 0 | 0 | **1** | 0 | 0 | 0 | 0 | 0 | 0 |
| sphere-6 | **1** | 0.97 | **1** | 0.79 | **1** | 0 | **1** | 0 | 0 | 0 | 0 | 0 | 0 | 0 | 0 | 0.67 | 0.7 | 0 | 0 | 0 |
| sphere-7 | **1** | **1** | **1** | **1** | **1** | 0 | **1** | 0 | 0.45 | 0 | 0 | 0 | 0 | 0 | 0.96 | 0.07 | **1** | 0 | 0 | **1** |
| sphere-8 | **1** | **1** | **1** | 0.99 | **1** | 0 | **1** | **1** | 0.23 | 0 | 0.02 | 0 | 0 | **1** | 0.98 | **1** | **1** | **1** | 0 | 0 |
| cuboid-1 | **1** | **1** | **1** | **1** | **1** | 0 | **1** | 0.13 | **1** | 0 | 0.38 | 0 | **1** | 0 | 0.72 | 0.13 | **1** | 0.4 | **1** | 0 |
| cuboid-2 | **1** | 0.91 | 0.99 | **1** | **1** | 0 | **1** | 0 | **1** | 0.06 | 0.13 | 0 | **1** | 0.1 | 0 | 0 | 0.79 | 0 | **1** | 0 |
| cuboid-3 | **1** | **1** | **1** | 0.92 | **1** | 0 | **1** | **1** | **1** | 0.98 | **1** | 0 | 0 | 0 | **1** | **1** | **1** | 0 | **1** | **1** |
| cuboid-4 | **1** | **1** | **1** | **1** | **1** | 0.12 | **1** | 0.13 | **1** | 0 | 0.87 | 0 | **1** | 0.1 | 0.88 | 0.93 | **1** | **1** | **1** | **1** |
| cuboid-5 | **1** | **1** | **1** | **1** | **1** | 0 | **1** | 0.13 | **1** | 0.13 | 0.96 | 0 | **1** | 0.2 | 0.99 | 0.73 | **1** | **1** | **1** | 0 |
| cuboid-6 | **1** | **1** | **1** | **1** | **1** | 0 | **1** | **1** | **1** | 0.93 | 0.83 | 0 | 0 | **1** | 0.93 | 0 | **1** | 0 | **1** | **1** |
| regular-1 | 0 | 0 | **1** | 0.04 | **1** | 0 | **1** | 0 | 0 | 0 | 0 | 0 | 0 | 0 | 0 | 0 | 0 | 0 | 0 | 0 |
| regular-2 | 0 | 0 | 0.58 | 0.01 | **0.7** | 0 | 0 | 0 | 0 | 0 | 0 | 0 | 0 | 0 | 0 | 0 | 0 | 0 | 0 | 0 |
| regular-3 | **1** | **1** | **1** | 0.96 | 0.87 | 0 | **1** | 0 | 0 | 0 | 0.17 | 0 | 0 | **1** | 0.33 | 0.73 | **1** | 0 | 0.85 | 0 |
| tetrahedron-1 | 0 | 0 | 0 | 0 | 0 | 0 | 0 | 0 | 0 | 0 | 0 | 0 | 0 | 0 | 0 | 0 | 0 | 0 | 0 | 0 |
| tetrahedron-2 | **1** | **1** | **1** | **1** | **1** | 0 | **1** | 0.13 | 0 | 0.5 | 0.45 | 0 | 0 | 0 | 0.98 | **1** | 0 | 0 | **1** | **1** |
| tetrahedron-3 | 0 | 0 | 0 | 0 | 0 | 0 | 0 | 0 | 0 | 0 | 0 | 0 | 0 | 0 | 0 | 0 | 0 | 0 | 0 | 0 |
| tetrahedron-4 | 0 | 0 | 0 | 0 | 0 | 0 | 0 | 0 | 0 | 0 | 0 | 0 | 0 | 0 | 0 | 0 | 0 | 0 | 0 | 0 |
| zero count | 19 | 23 | 15 | 25 | **14** | 67 | 26 | 46 | 45 | 46 | 34 | 68 | 55 | 49 | 43 | 47 | 48 | 59 | 36 | 56 |
| average | 68.75% | 59.31% | 64.86% | 47.11% | **72.92%** | 4.39% | 54.55% | 24.44% | 29.18% | 16.46% | 22.65% | 0.94% | 22.54% | 23.99% | 29.39% | 24.01% | 25.13% | 11.82% | 43.17% | 18.10% |
| average-easy | 98.89% | 99.54% | 99.96% | 99.11% | 99.54% | 11.14% | **100.00%** | 55.46% | 66.21% | 36.50% | 48.75% | 2.39% | 53.57% | 48.93% | 71.96% | 55.21% | 44.07% | 29.96% | 66.46% | 43.57% |
| average-medium | 82.61% | 60.22% | 73.30% | 24.48% | **94.61%** | 0.00% | 46.65% | 7.91% | 9.48% | 6.39% | 10.57% | 0.00% | 4.35% | 10.13% | 3.13% | 6.91% | 23.91% | 0.00% | 52.22% | 2.83% |
| average-hard | 10.60% | 1.95% | 6.00% | 0.35% | **10.70%** | 0.00% | 0.00% | 0.00% | 0.00% | 0.00% | 0.00% | 0.00% | 0.00% | 5.00% | 0.00% | 0.00% | 0.00% | 0.00% | 0.15% | 0.00% |

Table 7: result of data with constant, the discount means the recovery rate discount between no constant and with constant.

| baselines | 0-count | 0-recovery | 0.00001-count | 0.00001-recovery | 0.001-count | 0.001-recovery | discount |
|---|---|---|---|---|---|---|---|
| PSRN | 52 | 68.75% | 47 | 59.65% | 40 | 48.32% | 70.29% |
| Pysr | 48 | 59.31% | 44 | 53.51% | 44 | 49.31% | 83.13% |
| NGGP | 56 | 64.86% | 50 | 52.37% | 48 | 50.84% | 78.39% |
| UDSR | 46 | 47.11% | 43 | 37.56% | 43 | 35.56% | 75.47% |
| RSRM | 57 | 72.92% | 52 | 61.15% | 48 | 51.89% | 71.17% |
| KAN | 4 | 4.39% | 4 | 3.78% | 4 | 3.11% | 70.68% |
| BMS | 45 | 54.55% | 39 | 44.74% | 37 | 38.74% | 71.01% |
| PhySO | 25 | 24.44% | 25 | 22.90% | 23 | 21.81% | 89.23% |
| symindy | 26 | 29.18% | 20 | 25.37% | 18 | 21.34% | 73.11% |
| gplearn | 25 | 16.46% | 23 | 13.19% | 21 | 11.99% | 72.83% |
| deap | 37 | 22.65% | 30 | 17.85% | 27 | 17.54% | 77.43% |
| EQL | 3 | 0.94% | 2 | 0.83% | 2 | 0.67% | 70.94% |
| Sindy | 16 | 22.54% | 15 | 19.35% | 13 | 17.14% | 76.05% |
| SPL | 22 | 23.99% | 20 | 21.55% | 18 | 20.16% | 84.03% |
| E2E | 28 | 29.39% | 23 | 25.64% | 20 | 21.06% | 71.64% |
| TPSR | 24 | 24.01% | 14 | 19.06% | 10 | 14.85% | 61.84% |
| AIF | 23 | 25.13% | 18 | 21.87% | 17 | 15.60% | 62.10% |
| SNIP | 12 | 11.82% | 10 | 10.23% | 6 | 6.31% | 53.36% |
| DGSR | 35 | 43.17% | 29 | 34.60% | 28 | 30.41% | 70.44% |
| NSRS | 15 | 18.10% | 9 | 15.70% | 5 | 10.42% | 57.59% |

# F  EXTRA EXPERIMENTS

We have expanded our dataset to include four additional experiments concerning noise, speed, more baselines, and the introduction of constants.

Noise: We incorporated datasets with two levels of noise—1e-5 and 0.0001—to evaluate how well the models perform under noisy conditions. Unlike typical setups where only the target variable is affected by noise, we introduced noise equally to both the input variables and the target. This simulates a more realistic scenario where the measurement of both features and targets may be impacted by noise.

Speed: To assess the computational efficiency of each baseline, we measured the speed by averaging the results of 100 parallel runs for each category. Understanding the speed of each model is crucial as it allows us to create a Pareto front that balances the recovery rate against the computational time cost, providing a comprehensive view of model performance.

More Baselines: Recognizing the importance of robust comparison, we included additional baselines from the era of genetic programming. Specifically, we added Operon to our benchmarking table to evaluate its performance against other established methods.

Adding Constants: Since our dataset primarily comprises geometric equations where the only constant is $\pi$, we tested the ability of the symbolic regression models to handle constants by multiplying each dataset by a uniform constant ranging from 0 to 5. This test aims to assess each baseline's capability in accurately recovering symbolic expressions that incorporate constants.

The outcomes of these experiments are detailed below, illustrating how each model fares across these varied conditions and providing insights into their overall robustness and effectiveness in symbolic regression tasks.

from the result, all baselines suffer a lot from noise. And baselines with transformer pre-train module like SNIP, NSRS suffers most. And AIF RSRM also does not perform well due to their searching algorithm is not able to displace these noise.

As well, PhySO and pysr still have low discount due to their symbolic ability on physical dimensions. With the dimension , they can cut a lot of useless equation. Also, dimension is not affected through noise.

Table 8: the average time cost within each baselines within 71 datasets and 100 parallel runs.

| baselines | time cost(s) |
|---|---|
| PSRN | 270 |
| Pysr | 374 |
| NGGP | 2341 |
| UDSR | 3512 |
| RSRM | 2794 |
| KAN | 130 |
| BMS | 2371 |
| physo | 2098 |
| symindy | 478 |
| gplearn | 523 |
| deap | 476 |
| EQL | 1209 |
| Sindy | 0 |
| SPL | 1438 |
| E2E | 1 |
| TPSR | 3602 |
| AIF | 3475 |
| SNIP | 2975 |
| DGSR | 1097 |
| NSRS | 746 |

Next comes the speed test. In this test, we test each model's running speed through all 71 datasets. SIndy model, KAN model runs fast due to they are linear model. Also, end2end transformer model also runs fast for it only runs once and optimize its constant.

Then is the new baselines. Operon can reach 50.1% with 45 reachable, which is sightly below PySR within 127s average, but much more better than gplearn or deap.

Final is the constant learning:

We can conclude that PSRN can not good at handle with constant while the others can fit as well as before since the constant is only multiplies outside the equation.

Table 9: result of data with constant, the discount means the recovery rate discount between no constant and with constant.

| baselines | nonconstant-recovery | constant-recovery | discount |
|-----------|---------------------|-------------------|----------|
| PSRN | 68.75% | 51.21% | 74.49% |
| Pysr | 59.31% | 56.48% | 95.23% |
| NGGP | 64.86% | 59.78% | 92.17% |
| UDSR | 47.11% | 42.96% | 91.19% |
| RSRM | 72.92% | 63.98% | 87.74% |
| KAN | 4.39% | 3.97% | 90.49% |
| BMS | 54.55% | 47.24% | 86.60% |
| PhySO | 24.44% | 23.58% | 96.49% |
| symindy | 29.18% | 27.16% | 93.07% |
| gplearn | 16.46% | 15.82% | 96.09% |
| deap | 22.65% | 20.23% | 89.32% |
| EQL | 0.94% | 0.97% | 103.00% |
| Sindy | 22.54% | 22.00% | 97.60% |
| SPL | 23.99% | 22.45% | 93.56% |
| E2E | 29.39% | 26.13% | 88.91% |
| TPSR | 24.01% | 22.12% | 92.11% |
| AIF | 25.13% | 21.86% | 86.98% |
| SNIP | 11.82% | 10.85% | 91.81% |
| DGSR | 43.17% | 39.79% | 92.17% |
| NSRS | 18.10% | 16.84% | 93.05% |

