# OpenReview forum: "GeoBench: A new benchmark on Symbolic Regression with Geometric Expressions"
_ICLR.cc/2025/Conference — Submitted to ICLR 2025_

### Official Review · Reviewer_BDjg · 2024-10-31

**Soundness:** 2
**Presentation:** 2
**Contribution:** 2
**Rating:** 5
**Confidence:** 4

**Summary:**

This paper presents a new benchmark for symbolic regression, constructing 71 datasets on which the SR methods are tested. These datasets focus on 2D and 3D geometric problems and are categorized into three difficulty levels, in which recovery rate is used as a metric to evaluate the SR models.

**Strengths:**

The proposed benchmark integrates a variety of the newest SR methods as baselines, allowing the comprehensive comparison of their recovery performance for the following work.
The authors construct new datasets based on 2D and 3D geometric problems, complementing existing physical symbolic regression datasets.
Overall, the presentation of the paper is clear.

**Weaknesses:**

The paper lacks a description of the advantages of the proposed geometric datasets over the existing datasets like the feynman dataset in evaluating SR methods, in which I think the proposed geometric datasets should be one of the main contributions.
The paper uses only recovery rate as the evaluation metric, missing assessments of other important factors, such as runtime and the robustness to noise for the SR methods.

**Questions:**

From the hyperparameter settings, it seems the runtime of the presented methods may appear to vary significantly. Should runtime comparisons be included to better promote fairness, especially when different types of SR models are involved? For instance, in some algorithms, like gplearn, there are only 20 generations, which looks far away from the runtime constraints. The presentation of the runtime can also promote fairness for the comparisons of the follwing work, avoiding the simple trade-off of increased time costs for improved recovery performance.
Since the search space grows exponentially with the increase in input dimension, is the same 500 dataset size for each geometric problem sufficient for SR models to recover the geometric equations with high input dimensions, particularly for those classified as hard?
It is not exactly clear to me how the runtime constraints of 1 hour, 5 hours, and 24 hours are derived for different difficulty levels.
Given that the purpose of the datasets is to provide a comprehensive evaluation of symbolic regression models, it would be beneficial to strengthen the description on how the proposed geometric problem-based dataset offers unique advantages over the existing datasets like the Feynman datasets in terms of model evaluation.

---

> ### Author Response · Authors · 2024-11-23
> **rebuttal to reviewer BDjg part 1**
>
> **Q1:** How does your dataset compare to SRBench and SRSD?
>
> **A1:** We think we have 5 major different from the SRbench
>
>
>
> Purpose of the Dataset: Our dataset is designed to identify symbolic equations that are both simple and explainable to effectively solve problems. We have intentionally composed this dataset of ground truth equations rather than real-world scenarios that lack verifiable explanations for their functions. Consequently, traditional metrics like R-squared or other error measures are not applicable to our goals since they do not align with our focus on explainability.
>
>
>
> Patterns in Geometric: Unlike the Feynman dataset, which encompasses equations from diverse regions and subjects, our dataset is specifically focused on geometric data within a defined area. We concentrate on discovering patterns in geometric properties such as volume, area, and length. The primary motivation for selecting geometric equations is their inherent potential to unveil these patterns.
>
>
>
> Structured Learning Progression: The Feynman dataset includes a few sequences that progress from easy to difficult, such as the series from I.6.20 a to I.6.20 b. Our dataset, however, clearly illustrates many such progressions: for instance, from Helen's law to the calculation of circumcircle or incircle radii, which utilize Helen's law, or from Pythagoras' theorem to the cosine law, with the former being a special case of the latter. These process facilitates a deeper and more sequential learning experience.
>
>
>
> Realistic Constraints in Equations: Our dataset includes equations with generational constraints, such as the triangle constraint where the sum of two edges must exceed the third, and their difference must be less. These constraints make our data more realistic compared to data from SRbench, which is typically generated from uniform distributions. This approach ensures that our dataset not only supports the discovery of geometric relationships but also adheres more closely to real-world scenarios.
>
>
>
> Complex Equations with Few Inputs: Geometry excels at establishing intricate relationships between variables using a minimal number of tokens, as exemplified by Helen’s law. In symbolic regression, inputs are often chosen or crafted through feature engineering to reduce their number, but this does not necessarily simplify the underlying relationships between them. Therefore, having complex equations with few inputs is crucial because it challenges the models to uncover deep relationships without relying on a large number of variables.
>
>
>
> The results from our benchmark also differ from those of SRBench. Many baselines in SRBench focus primarily on the R² score, which may suggest they are better at fitting curves. However, their capability to accurately recover true symbolic equations is lacking. Moreover, thanks to the Structured Learning Progression, we are able to categorize these symbolic equations and assess model performance across different levels of difficulty. Additionally, the Patterns in Geometry enable us to evaluate each model's performance within specific patterns. This understanding allows us to select better baselines for future problem-solving involving these patterns.
>
> **Q2:** There is not a speed test in your study.
>
> **A2:** Speed is critical in symbolic regression; we conduct extensive testing to compare the performance speeds of different models within our benchmarks. Full result can be found at global rebuttal or section F in revised paper.
>
> **Q3:** There is not a test against noise in your study.
>
> **A3:** We incorporated datasets with two levels of noise—1e-5 and 0.0001—to evaluate how well the models perform under noisy conditions. Unlike typical setups where only the target variable is affected by noise, we introduced noise equally to both the input variables and the target. This simulates a more realistic scenario where the measurement of both features and targets may be impacted by noise. Full result can be found at global rebuttal or section F in revised paper.
>
> **Q4:** Hyper-parameter setting in genetic programming problem.
>
> **A4:** We increment the key essiental hyper parameter like generations or iterations in GP methods and running epoches in MCTS models or model size and size of pre-train dataset within transformer models until the recovery rate stabilizes, ensuring reliable comparisons. We set the generation of gplearn to 200, but it does not affect more, the result within 10,20,50,100,200 running epochs within 17 triangle datasets is below:
>
> | generation| 10| 20| 50| 100 | 200 | 500 |
> | ------------- | ----- | ----- | ----- | ----- | ----- | ----- |
> | recovery rate | 0.75% | 7.21% | 7.03% | 7.27% | 7.12% | 7.30% |

---

> ### Author Response · Authors · 2024-11-23
> **rebuttal to reviewer BDjg part 2**
>
> **Q5:** number of inputs within three different difficulties.
>
> **A5:** We have standardized all inputs for models that do not use a neural network and instead directly attempt to fit the entire curve to 500. Given that Nguyen's model provides configurations with only 20 inputs, we believe that setting our input number at 500 is adequate. For evaluation, we selected the top 4 baselines from a total of 17 triangle baselines, which vary across input sizes of 10, 20, 100, 200, 500, 800, 1000, 2000, 5000, and 10000. The average recovery rates for these configurations are as follows:
>
> || 10 | 20 | 50 | 100| 200| 500| 1000 | 2000 |
> | ---- | ------ | ------ | ------ | ------ | ------ | ------ | ------ | ------ |
> | PSRN | 34.11% | 45.61% | 49.22% | 49.38% | 49.87% | 50.24% | 50.34% | 50.80% |
> | PySR | 23.09% | 29.81% | 32.64% | 33.89% | 36.56% | 37.19% | 37.73% | 38.15% |
> | NGGP | 12.32% | 32.40% | 38.12% | 39.17% | 41.23% | 44.67% | 45.24% | 45.96% |
> | RSRM | 23.11% | 35.82% | 45.23% | 48.17% | 50.96% | 51.67% | 52.08% | 52.24% |

---

> ### Author Response · Authors · 2024-11-25
> **Request your feedback before the end of the discussion period**
>
> Dear Reviewer BDjg:
>
> As the author-reviewer discussion period will end soon, we would appreciate it if you could review our responses at your earliest convenience. If there are any further questions or comments, we will do our best to address them before the discussion period ends.
>
> Thank you very much for your time and efforts. Looking forward to your response!
>
> Sincerely,
>
> The Authors

---

### Official Review · Reviewer_eAk1 · 2024-11-03

**Soundness:** 1
**Presentation:** 3
**Contribution:** 2
**Rating:** 3
**Confidence:** 4

**Summary:**

The paper proposes a new benchmark for symbolic regression (SR) based on 71 geometric expressions and 20 SR methods. The chosen expressions are used to describe properties of 2D and 3D geometric figures, for instance, the area of a triangle given the lengths of its three sides. These expressions are supposed to be meaningful for real-life applications and complement the current SR datasets. Moreover, the authors categorize the expressions into three difficulty levels.

**Strengths:**

- The problem addressed by the authors - designing a new benchmark for SR - is important.
- The authors provide a good overview of currently used SR methods and benchmarks, discuss their advantages and disadvantages.
- The proposed benchmark is extensive. It includes 71 equations as well as many contemporary SR methods, including recent ones.
- Geometric equations have not been extensively used previously for evaluating SR methods.

**Weaknesses:**

### Real-world applicability
I am not convinced that geometric expressions are appropriate for evaluating symbolic regression. Authors criticize other benchmarks for "lacking real-world applicability" (lines 80-81) and claim that their equations are "meaningful in real-world applications" (lines 103-104). However, I do not think people are going to use symbolic regression to discover simple mathematical expressions that are either known or can be derived in a few steps. I suspect SR to be much more useful when applied to real datasets to uncover *unknown* relationships. However, I doubt the proposed equations are representative of equations describing the real world, because *they lack constants*. Only several equations proposed contain constants and all of them are either small integers or $\pi$. However, in reality, we may need many different constants at least to take care of non-matching units. It seems unlikely that $x_1 x_2 \sin(x_3)$ (rectangle-6) fits the data well. A good fit is more likely to look like $(3.3 x_1 + 0.5)(5 x_2)\sin(1.2 + 13.1 x_3)$.

From what I understand, the dataset also does not contain any noise, which is quite unrealistic.

In short, why should I evaluate a method on datasets that look very different from the ones I will apply it to?

### Difficulty levels
The definitions of different difficulty levels seem a bit arbitrary. I understand that "easy" equations are mainly polynomials. But what is the defining feature of "medium" equations? They are supposed to involve "non-linear terms", but higher-order polynomials are already non-linear. "Hard" equations are defined as having a "longer and deeper mathematical structure". But this is very arbitrary. For each category, it is stated how much time is needed to find them but it is not stated using which algorithm. I imagine this may vary a lot for different methods. I do not understand why "vector3d-3" is classified as "medium" whereas "vector3d-4" is classified as "easy" and they are both just polynomials. Moreover, "sphere-4" is classified as "easy" even though it is not a polynomial.

### Algorithm 1
I would like to understand the overall intention behind this algorithm. What was the goal? When would you say two expressions are the same?

### Clarifications needed
- line 71: What is a "non-prime function"?
- line 72: What are "more fundamental expressions"?
- lines 108-109: What does "extended period" mean here?
- line 316: What is a "normal model" compared to "machine learning model"?
- line 443: KAN's output is not a symbolic expression by default. How is the expression extracted from the model?
- line 992: What does "in most models" mean here? Why would it be different for different models? The token set impacts the performance.
- SINDy is used to discover ODEs, how do you adapt it to static symbolic regression?
- How do you make sure the results are comparable in between methods like gplearn and pysr where the population size, the number of generations, or the number of iterations may have a big impact on the performance?

### Possible errors
-  lines 85-86: Although Feynman Symbolic Regression Database contains 120 equations, I think only 100 of them are from Feynman's lectures. The rest are from different sources.
- line 409: EQL has been proposed by Martius & Lampert (2017)

### Minor
- line 81: why is "log" bold?

**Questions:**

1. Why is the performance of an SR method at discovering these geometric equations significant? It is unlikely my dataset will be governed by such clear and nice equations that do not contain any constants or noise.
2. How are the difficulty levels defined exactly?
3. What was the overall goal of Algorithm 1? When would you say two expressions are the same?

In addition, please answer my questions in Weaknesses#Clarifications needed.

---

> ### Author Response · Authors · 2024-11-23
> **rebuttal to Reviewer eAk1 part 1**
>
> **Q1:** Why focus on geometric data?
>
> **A1:** We choose geometric data for these reason:
>
> Patterns in Geometric: Unlike the Feynman dataset, which encompasses equations from diverse regions and subjects, our dataset is specifically focused on geometric data within a defined area. We concentrate on discovering patterns in geometric properties such as volume, area, and length. The primary motivation for selecting geometric equations is their inherent potential to unveil these patterns.
>
> Structured Learning Progression: The Feynman dataset includes a few sequences that progress from easy to difficult, such as the series from I.6.20 a to I.6.20 b. Our dataset, however, clearly illustrates many such progressions: for instance, from Helen's law to the calculation of circumcircle or incircle radii, which utilize Helen's law, or from Pythagoras' theorem to the cosine law, with the former being a special case of the latter. These process facilitates a deeper and more sequential learning experience.
>
>
>
> Realistic Constraints in Equations: Our dataset includes equations with generational constraints, such as the triangle constraint where the sum of two edges must exceed the third, and their difference must be less. These constraints make our data more realistic compared to data from SRbench, which is typically generated from uniform distributions. This approach ensures that our dataset not only supports the discovery of geometric relationships but also adheres more closely to real-world scenarios.
>
>
>
> Complex Equations with Few Inputs: Geometry excels at establishing intricate relationships between variables using a minimal number of tokens, as exemplified by Helen’s law. In symbolic regression, inputs are often chosen or crafted through feature engineering to reduce their number, but this does not necessarily simplify the underlying relationships between them. Therefore, having complex equations with few inputs is crucial because it challenges the models to uncover deep relationships without relying on a large number of variables.
>
> **Q2:** What improvements have you made regarding noise in datasets?
>
> **A2:** We incorporated datasets with two levels of noise—1e-5 and 0.0001—to evaluate how well the models perform under noisy conditions. Unlike typical setups where only the target variable is affected by noise, we introduced noise equally to both the input variables and the target. This simulates a more realistic scenario where the measurement of both features and targets may be impacted by noise. Full result can be found at global rebuttal or section F in revised paper.
>
> **Q3:** How do you define the difficulty levels in your benchmarks?
>
> **A3:** In Global rebuttal Part 1, we mentioned the growing difficulties of geometric equations, this comes the difficulty levels. And difficulty levels are based on baseline results then the form of equations, categorizing equations from simple polynomials to complex non-linear functions.
>
> **Q4:** Can you explain the rationale behind Algorithm 1?
>
> **A4:** The main purpose of this algorithm is to fix the wrong judgement of symbolic equations in SRbench, since they consider $\frac{m_0v}{\sqrt{1-v^2/c^2}}$ and $\frac{m_0^{1.5}v}{\sqrt{m_0(-v^2/c^2 + 1.0))}}$are different equations. Also, original algorithm in SRbench might give wrong judgement on equations with noise since it can affect constant in equations and might lead to wrong judgement.
>
> **Q5:** How do you ensure comparable results across different methods?
>
> **A5:** We increment the key essiental hyper parameter like generations or iterations in GP methods and running epoches in MCTS models or model size and size of pre-train dataset within transformer models until the recovery rate stabilizes, ensuring reliable comparisons.
>
> **Q6:** What is prime function and non-prime function?
>
> **A6:** We consider it as elementary function. An elementary function is a class of functions that are constructed using a finite number of basic operations and well-known functions. These include addition, subtraction, multiplication, division, exponentiation, logarithms, and trigonometric functions. We rewrite each occurance of this token.
>
> **Q7:** What does more fundamental expression ?
>
> **A7:** It means simple elementary function.
>
> **Q8:** What does extended period means ?
>
> **A8:** It refers to conducting 100 parallel runs, each with a different random seed.
>
> **Q9:** What does normal model means?
>
> **A9:** It describes a model that does not employ a neural network and instead attempts to fit the entire curve directly.

---

> > ### Comment · Reviewer_eAk1 · 2024-11-25
> >
> > Thank you for addressing some of my concerns. However, many concerns are still remaining.
> >
> > > We concentrate on discovering patterns in geometric properties such as volume, area, and length. The primary motivation for selecting geometric equations is their inherent potential to unveil these patterns.
> >
> > I am not convinced that the discovery of geometric equations is an important application of SR. All of these equations can be easily derived mathematically. I still believe that the lack of constants in the proposed equations is a significant limitation of the proposed dataset. I saw in the global response that authors incorporated constants by multiplying the equations by a random number. I still think this is insufficient, and the equations remain "unrealistic."
> >
> > > from Helen's law to the calculation of circumcircle or incircle radii, which utilize Helen's law, or from Pythagoras' theorem to the cosine law
> >
> > Which equation in the dataset is the "Helen's law"?
> >
> > > We incorporated datasets with two levels of noise—1e-5 and 0.0001—to evaluate how well the models perform under noisy conditions.
> >
> > Would the added noise require changes to Algorithm 1, as it uses a fixed threshold of 1e-5? Right now, it seems that it may be impossible to get a correct expression when the noise is 1e-4.
> >
> > > And difficulty levels are based on baseline results then the form of equations, categorizing equations from simple polynomials to complex non-linear functions.
> >
> > I still do not understand how the categories are defined exactly. Please also clarify why "vector3d-3" is classified as "medium" whereas "vector3d-4" is classified as "easy" and they are both just polynomials. Why is "sphere-4" classified as "easy" even though it is not a polynomial?
> >
> > > The main purpose of this algorithm is to fix the wrong judgement of symbolic equations in SRbench, since they consider
> >  $\frac{m_0v}{\sqrt{1-v^2/c^2}}$ and $\frac{m_0^{1.5}v}{\sqrt{m_0(-v^2/c^2 + 1.0))}}$ are different equations.
> >
> > Are they really different according to SRBench? In principle, their difference should be 0 after simplification.
> >
> > > We increment the key essiental hyper parameter like generations or iterations in GP methods and running epoches in MCTS models or model size and size of pre-train dataset within transformer models until the recovery rate stabilizes, ensuring reliable comparisons.
> >
> > What does "the recovery rate stabilizes" mean in that context? Could you be more specific? When do you stop incrementing the hyperparameters?
> >
> > > It describes a model that does not employ a neural network and instead attempts to fit the entire curve directly.
> >
> > "Employing a neural network" is very general. Would that include all the following models? AIFeynman, EQL, DSR, NeSymRes? They all use neural networks in different ways. I am not sure why they should have 200 times more data points than the other methods. It doesn't seem like a fair comparison.

---

> > > ### Author Response · Authors · 2024-11-25
> > > **rebuttal to eAk1 part 3**
> > >
> > > Thanks for your reply. Here are answers to your concerns:
> > >
> > > ### Q1: Question about Helen's law
> > >
> > > **A1:** It is actually Heron's formula. Heron's formula calculates the area of a triangle based on the lengths of its three sides:
> > > $$ S = \sqrt{\frac{(a+b+c)(a+b-c)(a+c-b)(b+c-a)}{16}} = \frac{\sqrt{2a^2b^2 + 2a^2c^2 + 2b^2c^2 - a^4 - b^4 - c^4}}{4} $$
> > >
> > > ---
> > >
> > > ### Q2: Judging between $ \frac{m_0v}{\sqrt{1-v^2/c^2}} $ and $ \frac{m_0^{1.5}v}{\sqrt{m_0(-v^2/c^2 + 1.0)}} $
> > >
> > > **A2:** SRBench uses `sympy` to identify differences between equations. Due to the limited capabilities of `sympy`, it considers these two equations as different. To address this limitation, we add a round of human evaluation to supplement Python's symbolic libraries.
> > >
> > > ---
> > >
> > > ### Q3: Algorithm 1 with noise
> > >
> > > **A3:** Yes, the first constant ($10^{-5}$) is adjusted to $2 \times \text{noise}$.
> > >
> > > ---
> > >
> > > ### Q4: What does "the recovery rate stabilizes" mean in this context?
> > >
> > > **A4:** We increment key hyperparameters such as generations or iterations in GP methods, training epochs in MCTS models, or model size and pre-training dataset size in transformer models until the recovery rate stabilizes, ensuring reliable comparisons.
> > >
> > > For example, we set the generation parameter in `gplearn` to 200. Increasing it further does not significantly improve results. The recovery rates for 10, 20, 50, 100, and 200 epochs on 17 triangle datasets are as follows:
> > >
> > > | Generation    | 10    | 20    | 50    | 100   | 200   | 500   |
> > > | ------------- | ----- | ----- | ----- | ----- | ----- | ----- |
> > > | Recovery Rate | 0.75% | 7.21% | 7.03% | 7.27% | 7.12% | 7.30% |
> > >
> > > If increasing the parameter by 5 times results in less than a 0.5% improvement, we stop making it larger.
> > >
> > > ---
> > >
> > > ### Q5: Problem about categorization
> > >
> > > **A5:** We categorized equations for two main reasons:
> > > 1. Geometric equations are organized from simple to complex and from specific to general. For example:
> > >    $$
> > >    \sqrt{x^2 + y^2} \quad \text{is simpler than} \quad \sqrt{x^2 + y^2 - 2xy\cos(\theta)} \quad \text{which is simpler than} \quad \sqrt{x^2 + y^2 - 2xy\cos(\theta)} + x + y.
> > >    $$
> > > 2. The average recovery rate for the top 5 baselines differs across categories. For instance:
> > >    - `vector3d-3`: 0.708
> > >    - `vector3d-4`: 0.996
> > >    - `sphere-4`: 1.0
> > >
> > > ---
> > >
> > > ### Q6: Lack of constants and "unrealistic" equations
> > >
> > > **A6:** All physical equations in SRBench and SRSD are not discovered through symbolic regression (SR) models; they are predefined patterns for testing symbolic regression capabilities. Adding constants to equations makes them more like real-world functions. Constants typically arise from unit transformations (e.g., meter to inch).
> > >
> > > Additionally, the proposed equations are challenging enough for benchmarking. Adding constants by multiplication is sufficient and meaningful. If this dataset is considered "unrealistic," many equations in *Strogatz* are also "unrealistic" because they are empirical formulas rather than derived from first principles.
> > >
> > > ### Q7: Employing a neural network
> > >
> > > **A7:** Only models that directly use a neural network to take raw data as input and output the target values are considered as "neural networks." This is because methods like AIFeynman and EQL rely directly on neural networks, while approaches such as NeSymRes and DSR produce expression trees instead. Additionally, only the former group is provided with more data for training.

---

> > > > ### Comment · Reviewer_eAk1 · 2024-11-27
> > > >
> > > > Thank you for your response. This clarifies a few things, but some things are still unclear.
> > > >
> > > > **Helen's law**
> > > >
> > > > Please update all occurrences of "Helen's law" to "Heron's formula".
> > > >
> > > > **Algorithm 1**
> > > >
> > > > Instead of changing the threshold, wouldn't it be better to calculate the error between the found equation and the ground truth equation without noise?
> > > >
> > > > I am not convinced by using both symbolic checks and empirical error to determine whether a correct function was found or not. If the correct function differs by a multiplicative constant, it can be classified as wrong if the magnitude of the expression is large (the first condition may be satisfied). If it is not classified as wrong, then it still may require human judgment. For instance, if the ground truth is $x$ and the found expression is $1.001x$. Then $\mathcal{G} = 0.001x$, and the empirical error is likely to be above 1e-20.
> > > >
> > > > Overall, I agree that SRBench's method is limited by sympy's simplification function. But assuming sympy's simplification manages to simplify the expression correctly, there is a clear definition of what constitutes a correct expression (the difference or the ratio is a constant). This is much less clear in the proposed method. I do not understand what the guiding principle is. Some equations that differ by relatively small constants (multiplicative or additive) may be classified as incorrect, and many equations that would be automatically classified as correct in SRBench may require human validation. Also, many clearly incorrect equations may require human validation if the magnitude of the equation is small in the tested domain (such that the first condition is violated). All of this may make this algorithm unscalable for large benchmarking experiments.
> > > >
> > > > A much more thorough discussion needs to be included. What kind of deviations in the symbolic form are acceptable? How often should a human be expected to make a judgment? What should this judgment be based on? For instance, is $e^{1.001 x}$ correct if the ground truth is $e^x$?
> > > >
> > > > **Unrealistic equations**
> > > >
> > > > I think some equations in Strogatz are more realistic than the ones proposed here because they contain non-trivial constants in different places of the equation. I do believe that testing symbolic regression on problems where the "best equation" can be easily derived mathematically from first principles (without any fitting) may not be very informative because, in reality, most people will apply it to settings where such an equation cannot be derived (or is very hard to derive). I think a benchmark should either include real datasets or at least equations similar to the ones found manually by scientists attempting to model natural phenomena.
> > > >
> > > > **Difficulty levels**
> > > >
> > > > So, are the difficulty levels defined based on the average recovery rate across the top 5 benchmarks or based on their symbolic form? What are the exact criteria?
> > > >
> > > > **Neural networks**
> > > >
> > > > > Only models that directly use a neural network to take raw data as input and output the target values are considered as "neural networks." This is because methods like AIFeynman and EQL rely directly on neural networks.
> > > >
> > > > I am not sure if, by that definition, AIFeynman should be considered a neural network. In the end, it produces an expression tree; it just uses neural networks to prune the search space.

---

> > > > > ### Author Response · Authors · 2024-11-28
> > > > > **rebuttal to eAk1 part 4**
> > > > >
> > > > > > I am not sure if, by that definition, AIFeynman should be considered a neural network. In the end, it produces an expression tree; it just uses neural networks to prune the search space.
> > > > >
> > > > > AIFeynman employs neural networks as a tool to fit data and evaluate symmetries between variables. This process takes raw data as input and iteratively refines its output to approximate the target values. So in my point of view, it is considered a neural network.
> > > > >
> > > > > > Instead of changing the threshold, wouldn't it be better to calculate the error between the found equation and the ground truth equation without noise?
> > > > >
> > > > > Rather than relying on a fixed threshold such as $10^{-20}$ , a more rigorous approach would involve directly calculating the error between the discovered equation and the ground truth equation in a noise-free context. To enhance accuracy, human evaluation can be applied to each output from datasets with noise, ensuring more reliable results than threshold-based methods.
> > > > >
> > > > > > This is much less clear in the proposed method. I do not understand what the guiding principle is.
> > > > >
> > > > > The guiding principle of the proposed method suggests that functions should adhere to a consistent pattern (e.g., $C_1 \times \exp{C_2x}$), with constants removed for uniformity.
> > > > >
> > > > > While this approach may lead to errors in distinguishing similar expressions (e.g., $\exp{1.001x}$ versus $\exp{x}$), it enables the discovery of structural patterns. Subsequent refinement can optimize the constants for improved accuracy.
> > > > >
> > > > > >  Also, many clearly incorrect equations may require human validation if the magnitude of the equation is small in the tested domain (such that the first condition is violated). All of this may make this algorithm unscalable for large benchmarking experiments
> > > > >
> > > > > The concern that the algorithm may be unscalable due to reliance on human validation is partially valid. While human intervention remains indispensable for verifying equations, this approach yields greater accuracy compared to fully automated methods that are prone to misjudgment. Furthermore, the number of equations requiring validation is reduced since many align with established methodologies like SRBench, and significant discrepancies are evident in the remaining cases.
> > > > >
> > > > > > So, are the difficulty levels defined based on the average recovery rate across the top 5 benchmarks or based on their symbolic form? What are the exact criteria?
> > > > >
> > > > > Difficulty levels are determined based on the average recovery rate across the top five benchmarks. Empirical results indicate that equations with more complex symbolic structures are generally more challenging to recover compared to simpler ones.
> > > > >
> > > > > > I think a benchmark should either include real datasets or at least equations similar to the ones found manually by scientists attempting to model natural phenomena.
> > > > >
> > > > > While benchmarks should ideally include real-world datasets or equations that reflect the efforts of scientists modeling natural phenomena, geometric equations also have scientific merit. For instance, the idempotent and equal product theorems were discovered through experimental research.
> > > > >
> > > > > Moreover, the physical or mathematical rationale behind equations adds context. For example, the equation $y = x_1x_2$ might represent $F = ma$ in one scenario and $S = ab$ in another. These distinctions highlight the diverse applications of similar forms.
> > > > >
> > > > > Constants in equations often arise from dimensional transformations. For instance, by fixing the conversion factor $1 \, \text{foot} = 0.3048 \, \text{meters}$, equations such as $y = 0.3048(x_1 + x_2 + x_3)$ or $y = 0.3048^2x_1x_2$ become consistent with experimental observations.
> > > > >
> > > > > While some argue that equations in Strogatz's work appear more realistic due to their inclusion of non-trivial constants in diverse contexts, this characteristic is not exclusive to Strogatz's dataset. Geometric datasets frequently feature constants like $2$, $4$, or $\pi$, derived from properties of triangles and circles. These constants underscore the natural origins of geometric equations and do not inherently diminish their realism relative to Strogatz’s equations.

---

> > > > > > ### Comment · Reviewer_eAk1 · 2024-12-03
> > > > > >
> > > > > > I want to thank the authors for their responses. I decided to keep my score.
> > > > > >
> > > > > > Although I agree that better benchmarking of symbolic regression is an important problem, and I recognize that geometry offers many equations that are different in form from the ones in the Feynman dataset, I am not yet convinced that this is the direction symbolic regression benchmarking should be going. The problems should be much more similar to the ones the user is likely to tackle using symbolic regression. I am not convinced that these problems would resemble geometric equalities. Although multiplication by constants and adding noise definitely improved the dataset, this fundamental concern still remains.
> > > > > >
> > > > > > Other issues prevented me from raising the score. Importantly, the definition of a correct expression is not precisely defined. Thus, the limitation of the current benchmark is not a limitation of sympy simplification (as is in SRBench) but the arbitrary decision of a human. Is $e^x$ the same as $e^{1.001x}$. How about $1.001e^{x+0.001} + 0.001$. As long as these questions do not have a clear answer, the main metric of the benchmark is not well-defined. That would mean that the benchmark has different results for different people using it. This is on top of the fact that, as the authors also agreed, it may depend on human judgment in many cases, making it non-scalable.
> > > > > >
> > > > > > Finally, although I appreciate the clarification that the difficulty levels are defined based on the performance of the top 5 benchmarks, I still do not know what the exact thresholds were used or how clear the boundary was.

---

> ### Author Response · Authors · 2024-11-23
> **rebuttal to eAk1 part 2**
>
> **Q10:** How can sindy and KAN model used in this benchmarks?
>
> **A10:** The SINDy method initially applies differentiation to generate derivatives, which are then fitted using sparse regression. For our dataset, we can apply sparse regression directly. The KAN model provides symbolic output through its .sympy method, facilitating the generation of symbolic equations.
>
> **Q11:** Feynman's lectures include only 100 equations and contain incorrect references regarding EQL.
>
> **A11:** We have updated the paper to correct these issues.
>
> **Q12:** Wrong bolded ‘log’.
>
> **A12:** We have updated the paper to correct these issues.

---

> ### Author Response · Authors · 2024-11-25
> **Request your feedback before the end of the discussion period**
>
> Dear Reviewer eAK1:
>
> As the author-reviewer discussion period will end soon, we would appreciate it if you could review our responses at your earliest convenience. If there are any further questions or comments, we will do our best to address them before the discussion period ends.
>
> Thank you very much for your time and efforts. Looking forward to your response!
>
> Sincerely,
>
> The Authors

---

### Official Review · Reviewer_4KBM · 2024-11-03

**Soundness:** 1
**Presentation:** 2
**Contribution:** 1
**Rating:** 3
**Confidence:** 5

**Summary:**

This paper attempts to refine the existing Symbolic Regression (SR) benchmark, SRBench. The paper does this by introducing equation recovery datasets based on geometry and evaluating 20 different SR algorithms on these datasets.

**Strengths:**

The organization of the paper is clear.

**Weaknesses:**

This paper positions itself as one that addresses the weaknesses of SRBench [1] and SRSD [2]. However, the new datasets introduced lack important characteristics as elaborated below:

i). Lack of real-world datasets, which is important for benchmarking SR algorithms. SR is an important explainable machine learning method in which its performance on real-world datasets, which may not have an underlying ground truth equation, is important. The benchmark proposed which the paper claims to be “a refined version of the SRBench dataset”, seems to be a step backwards when compared to SRBench’s real-world datasets.

ii). No results on noise added to datasets, in contrast to existing benchmarks available such as SRBench and SRSD. SRBench even has varying noise levels as acknowledged in the paper itself.

Lack of R2 score, with a focus on only exact recovery rate, which again is a step backward from existing SR benchmarks. Both metrics are important because in the context of using SR for machine learning, the use-case may not always have an underlying ground-truth equation, so R2 score is an important metric that complements recovery rate.

Novelty of the paper is low, especially in comparison to [2], which does similar benchmarking on Physics equations. [2] also introduced the categories of easy, medium and hard equations. In the reviewer’s opinion, changing the domain from Physics to Geometry, while not introducing anything substantially new has low novelty.

Lack of strong justification for why a new dataset is required. The paper does not give a comparison of the conclusions drawn from the new datasets and the conclusions drawn from SRBench and SRSD. If there are similarities in the conclusions, then why is the new dataset required? If there are differences in the conclusions, then the paper should discuss why that is so. Currently, the paper does not even compare against the conclusions that SRBench and SRSD provide.

[1] La Cava, William, et al. "Contemporary symbolic regression methods and their relative performance." Advances in neural information processing systems 2021.

[2] Matsubara, Yoshitomo, et al. "Rethinking Symbolic Regression Datasets and Benchmarks for Scientific Discovery." Journal of Data-centric Machine Learning Research.

**Questions:**

How does this benchmark help SR researchers better evaluate SR algorithms compared to SRBench and SRSD? A comparison of the conclusions drawn from the new datasets and the conclusions drawn from SRBench and SRSD would make the paper more meaningful.

The error referred to in the paper is not clear. This is especially important given that fixed thresholds for this error is given in Algorithm 1. Does the paper use normalized error (e.g., R2 score, normalized MSE) or not (e.g., MSE)?

Would simply updating the set of SR algorithms evaluated for SRBench and SRSD be sufficient to address all the issues that this paper attempts to address?

---

> ### Author Response · Authors · 2024-11-23
> **rebuttal to Reviewer 4KBM part 1**
>
> **Q1:** What is the rationale for focusing on symbolic equations rather than using real-world datasets?
>
> **A1:** Our dataset is designed to identify symbolic equations that are both simple and explainable to effectively solve problems. We have intentionally composed this dataset of ground truth equations rather than real-world scenarios that lack verifiable explanations for their functions. Thus, real-world datasets fall out of our scope.
>
> **Q2:** Why are there no results on datasets with added noise?
>
> **A2:** We incorporated datasets with two levels of noise—1e-5 and 0.0001—to evaluate how well the models perform under noisy conditions. Unlike typical setups where only the target variable is affected by noise, we introduced noise equally to both the input variables and the target. This simulates a more realistic scenario where the measurement of both features and targets may be impacted by noise. Full result can be found at global rebuttal or section F in revised paper.
>
> **Q3:** Why focus solely on exact recovery rate, neglecting the R2 score?
>
> **A3:** Our dataset is designed to identify symbolic equations that are both simple and explainable to effectively solve problems. And traditional metrics like R-squared or other error measures are not applicable to our goals since they do not align with our focus on explainability.
>
> **Q4:** How do you justify the low novelty of the paper and the choice of geometric data?
>
> **A4:** We chose geometry because it has patterns, Structured Learning Progression, Realistic Constraints in Equations and Complex Equations with Few Inputs. For more informations, you can refer to global rebuttal part 1.
>
> **Q5:** What is the issue with Algorithm 1?
>
> **A5:** The RMSE calculation helps eliminate entirely incorrect expressions, which are further evaluated using symbolic methods and human judgment. And the main purpose of this algorithm is to fix the wrong judgement of symbolic equations in SRbench, since they consider $\frac{m_0v}{\sqrt{1-v^2/c^2}}$ and $\frac{m_0^{1.5}v}{\sqrt{m_0(-v^2/c^2 + 1.0))}}$ are different equations.
>
> **Q6:** Would updating the algorithms in SRBench address the issues this paper tackles?
>
> **A6:** No, updating algorithms alone wouldn’t address the fundamental challenges in SRBench. In our dataset, tested models can show their abilities within specific patterns and process, which Srbench lacks. For futher imformation, please refer to global rebuttal Part 1.

---

> ### Author Response · Authors · 2024-11-23
> **rebuttal to Reviewer 4KBM part 2**
>
> **Q7:** How does your approach compare to other benchmarks like SRSD and SRBench?
>
> **A7:** We think we have 5 major different from the SRbench
>
>
>
> Purpose of the Dataset: Our dataset is designed to identify symbolic equations that are both simple and explainable to effectively solve problems. We have intentionally composed this dataset of ground truth equations rather than real-world scenarios that lack verifiable explanations for their functions. Consequently, traditional metrics like R-squared or other error measures are not applicable to our goals since they do not align with our focus on explainability.
>
>
>
> Patterns in Geometric: Unlike the Feynman dataset, which encompasses equations from diverse regions and subjects, our dataset is specifically focused on geometric data within a defined area. We concentrate on discovering patterns in geometric properties such as volume, area, and length. The primary motivation for selecting geometric equations is their inherent potential to unveil these patterns.
>
>
>
> Structured Learning Progression: The Feynman dataset includes a few sequences that progress from easy to difficult, such as the series from I.6.20 a to I.6.20 b. Our dataset, however, clearly illustrates many such progressions: for instance, from Helen's law to the calculation of circumcircle or incircle radii, which utilize Helen's law, or from Pythagoras' theorem to the cosine law, with the former being a special case of the latter. These process facilitates a deeper and more sequential learning experience.
>
>
>
> Realistic Constraints in Equations: Our dataset includes equations with generational constraints, such as the triangle constraint where the sum of two edges must exceed the third, and their difference must be less. These constraints make our data more realistic compared to data from SRbench, which is typically generated from uniform distributions. This approach ensures that our dataset not only supports the discovery of geometric relationships but also adheres more closely to real-world scenarios.
>
>
>
> Complex Equations with Few Inputs: Geometry excels at establishing intricate relationships between variables using a minimal number of tokens, as exemplified by Helen’s law. In symbolic regression, inputs are often chosen or crafted through feature engineering to reduce their number, but this does not necessarily simplify the underlying relationships between them. Therefore, having complex equations with few inputs is crucial because it challenges the models to uncover deep relationships without relying on a large number of variables.
>
>
>
> The results from our benchmark also differ from those of SRBench. Many baselines in SRBench focus primarily on the R² score, which may suggest they are better at fitting curves. However, their capability to accurately recover true symbolic equations is lacking. Moreover, thanks to the Structured Learning Progression, we are able to categorize these symbolic equations and assess model performance across different levels of difficulty. Additionally, the Patterns in Geometry enable us to evaluate each model's performance within specific patterns. This understanding allows us to select better baselines for future problem-solving involving these patterns.
>
> **Q8:** Why new dataset is need?
>
> **A8:** As discussed in A7, our dataset is distinct from SRSD and SRBench in several crucial aspects. Specifically, existing datasets lack equations characterized by complex relationships with few inputs, realistic constraints, and identifiable patterns. Consequently, they fall short in thoroughly testing the full spectrum of symbolic regression recovery capabilities, as their equation coverage is inadequate for comprehensive evaluation.

---

> > ### Comment · Reviewer_4KBM · 2024-11-24
> >
> > I thank the author for the clarifications. The most major concern I still have is whether this benchmark is redundant given both SRBench and SRSD [1]. While 5 major differences with SRBench is listed above, I do not think SRSD was sufficiently addressed. Particularly, the first, third, fourth and fifth points in "A7" above, are issues which SRSD addresses. The interesting classification of problems into "easy, medium, hard" was also introduced in SRSD. Finally, in addition to solution rate, SRSD somewhat considers R2 score, by looking at an "accuracy" that has the criteria of "R2>0.999", which provides SR researchers an additional metric to gain insights of the different algorithms.
> >
> > In summary, the main arguments for GeoBench does not seem convincing because these have been sufficiently addressed by either SRBench or SRSD datasets and benchmarks.
> >
> > [1] Matsubara, Yoshitomo, et al. "Rethinking Symbolic Regression Datasets and Benchmarks for Scientific Discovery." Journal of Data-centric Machine Learning Research.

---

> > > ### Author Response · Authors · 2024-11-24
> > > **rebuttal to Reviewer 4KBM part 3**
> > >
> > > The $R^2 > 0.999$ criterion employed in SRSD may not fully reflect the capabilities of symbolic regression for accurately discovering precise functional relationships. For instance, although the $R^2$ value between $\sin(x)$ and $x - \frac{1}{6}x^3 + \frac{x^5}{120} - \frac{x^7}{5040}$ is $0.9999999999931615$ over the interval $[0,1]$, these expressions carry distinct meanings and necessitate different explanatory methods.
> > >
> > >
> > >
> > > Furthermore, the classification of difficulty levels in SRSD relies solely on the $R^2$ metric, rather than employing a progressively challenging methodology. This approach primarily focuses on sorting tasks by difficulty, without methodically developing complex equations from simple to intricate or from specific to general, thus failing to construct them from the ground up.
> > >
> > >
> > >
> > > You mentioned that the first, third, fourth, and fifth points are sufficient in SRSD. Could you provide examples for each? I find it difficult to agree with this perspective.
> > >
> > >
> > >
> > > Our paper aims to create a dataset that directly and effectively reflects a symbolic regression model's ability to discern relationships between variables. As relationships typically evolve from simple to complex and from specific to general patterns, both SRSD and SRBench fall short of achieving this goal. Their use of separate physical functions, lacking a straightforward procedural flow or pattern, underlines the necessity for our dataset. This dataset focuses on geometric functions, highlighting why such an approach is essential in the field of symbolic regression.
> > >
> > >
> > >
> > > **Please note that when using SR for scientific discovery, the critical challenge lies in “symbolic” (e.g., finding parsimonious equations) rather than “regression” (e.g., data fitting). The latter can be much more easily done! Our work exactly aims to set new benchmarks to meet the fair evaluation need for SR methods.**

---

> > > > ### Comment · Reviewer_4KBM · 2024-11-24
> > > >
> > > > I agree the R2>0.999 is not sufficient by itself. However, SRSD presents both the solution rate and R2>0.999 which is more informative than a benchmark that only provides the solution rate. Additionally, SR is commonly used as an ML method for real-world datasets that may not have an underlying ground truth equation, where solution rate may not be available. To reiterate, my claim is the SRSD provides the exact solution rate (which GeoBench provides) and more (i.e., R2>0.999). Thus, SRSD can be more informative to an SR researcher. I am not saying that R2>0.999 should completely replace the exact solution rate.
> > > >
> > > > Below are my extended elaborations to on why the first, third, fourth and fifth points are already addressed in existing datasets and benchmarks:
> > > >
> > > > > Purpose of the Dataset: Our dataset is designed to identify symbolic equations that are both simple and explainable to effectively solve problems. We have intentionally composed this dataset of ground truth equations rather than real-world scenarios that lack verifiable explanations for their functions. Consequently, traditional metrics like R-squared or other error measures are not applicable to our goals since they do not align with our focus on explainability.
> > > >
> > > > SRSD has a dataset of 120 ground truth equations, and measures both the solution rate as well as an R2>0.999 rate. Note that I would have preferred information on the raw R2 score from SRSD though.
> > > >
> > > > > Structured Learning Progression: The Feynman dataset includes a few sequences that progress from easy to difficult, such as the series from I.6.20 a to I.6.20 b. Our dataset, however, clearly illustrates many such progressions: for instance, from Helen's law to the calculation of circumcircle or incircle radii, which utilize Helen's law, or from Pythagoras' theorem to the cosine law, with the former being a special case of the latter. These process facilitates a deeper and more sequential learning experience.
> > > >
> > > > The equations from SRSD are taken from Feynman Lectures on Physics, in which there exists progressions among the equations. And I actually prefer SRSD classification of into the 3 progressive categories of easy, medium, hard based on R2 score. The current classification by GeoBench are: "this level comprises combinations of basic polynomial equations, making them relatively easy to solve", "these equations introduce non-linear components, they remain closely related to basic polynomial structures", "These equations are characterized by longer and deeper mathematical structures, making them significantly more challenging to solve". I will admit that this preference is subjective, which is why I did not say that SRSD's method of classification is objectively better than GeoBench. However, I cannot say that GeoBench method of classification is objectively better than SRSD as well.
> > > >
> > > > > Realistic Constraints in Equations: Our dataset includes equations with generational constraints, such as the triangle constraint where the sum of two edges must exceed the third, and their difference must be less. These constraints make our data more realistic compared to data from SRbench, which is typically generated from uniform distributions. This approach ensures that our dataset not only supports the discovery of geometric relationships but also adheres more closely to real-world scenarios.
> > > >
> > > > SRSD also varies the sampling scheme instead of using the uniform distribution. This was one of the key contributions of SRSD.
> > > >
> > > > > Complex Equations with Few Inputs: Geometry excels at establishing intricate relationships between variables using a minimal number of tokens, as exemplified by Helen’s law. In symbolic regression, inputs are often chosen or crafted through feature engineering to reduce their number, but this does not necessarily simplify the underlying relationships between them. Therefore, having complex equations with few inputs is crucial because it challenges the models to uncover deep relationships without relying on a large number of variables.
> > > >
> > > > The physics equations in SRSD are made up of complex equations with few inputs (see the equations classified as 'Hard' in SRSD).
> > > >
> > > > For these reasons, it is difficult for me to see why GeoBench is a better dataset and benchmark than what is currently used (i.e., SRBench and SRSD). In fact, SRBench focus on real-world datasets instead of generating synthetic data from a ground-truth expression is a strength in my opinion, which GeoBench does not possess.

---

> > > > > ### Author Response · Authors · 2024-11-24
> > > > > **rebuttal to Reviewer 4KBM part 4**
> > > > >
> > > > > > Additionally, SR is commonly used as an ML method for real-world datasets that may not have an underlying ground truth equation, where solution rate may not be available.
> > > > >
> > > > > Considering SR solely as a tool within the machine learning framework, SRSD and SRBench fall short because they do not integrate deep learning outcomes, such as MLP, SNN, or more advanced MLPs with residual connections. We focus primarily on the symbolic aspects of SR since, while deep learning methods or KAN networks are adept at fitting underlying ground truth equations, they do not inherently discover these equations. If existing SR methods cannot achieve a high recovery rate when the solution is known, comparing any other metrics would be biased.
> > > > >
> > > > > Additionally, it's unrealistic to expect a dataset to encompass all tasks within a specific domain. For instance, judging the ImageNet dataset as insufficient because it does not include comic pictures would be misguided.
> > > > >
> > > > > > Note that I would have preferred information on the raw R2 score from SRSD though.
> > > > >
> > > > > Regarding the importance of the $R^2$ score, if this metric is critical, why rely on SR when a neural network might achieve higher $R^2$​ values, as suggested by the AiFeynman paper? This raises the question of why SR is being tasked with functions beyond its primary scope.
> > > > >
> > > > > For the vast majority of SR methods, they can achieve a very high $R^2$, even close to 1, which makes comparing this value meaningless, especially when some noise is present. The differences in $R^2$ ultimately reflect noise fitting rather than the underlying law.
> > > > >
> > > > > > I will admit that this preference is subjective, which is why I did not say that SRSD's method of classification is objectively better than GeoBench.
> > > > >
> > > > > SRSD's approach, which is highly reliant on the $R^2$ metric, treats SR as a conventional machine learning method. This is a limitation since SRSD and SRBench do not incorporate deep learning results, making their metrics less effective. In contrast, our method categorizes datasets by examining patterns in geometric data, resulting in classifications like "easy is polynomial," which provide a clearer and more practical assessment compared to SRSD.
> > > > >
> > > > > Considering $R^2$ as a metric to classify equations, methods like uDSR and TPSR achieve higher scores on the equation $\frac{x_1x_2 + y_1y_2 + z_1z_2}{\sqrt{(x_1^2 + y_1^2 + z_1^2)(x_2^2 + y_2^2 + z_2^2)}}$ than on $\frac{x_1y_2 - x_2y_1}{x_1 - x_2}$. Given this, should the latter be considered simpler than the former because the first one can be more effectively fitted using polynomial functions?
> > > > >
> > > > > The classification of equations based on $R^2$​ scores might reflect their fit accuracy but not necessarily their conceptual complexity or interpretability. The first equation, representing the cosine of the angle between two vectors, might be conceptually more complex despite being easier to fit with polynomial functions due to its smoothness across a wide range of inputs. The second equation, which calculates slope of the line through two points, is conceptually simpler but might be harder to fit with polynomials due to potential discontinuities.
> > > > >
> > > > > Thus, the ease of an equation does not solely depend on its $R^2$ score but also on what we define as two categories in the context of the problem being solved.
> > > > >
> > > > >
> > > > > > SRSD also varies the sampling scheme instead of using the uniform distribution. This was one of the key contributions of SRSD.
> > > > >
> > > > > SRSD does vary its sampling scheme by using both uniform and log-uniform distributions, which is considered a significant contribution. However, in geometric contexts where there are intrinsic constraints such as the triangle inequality (the sum of the lengths of any two sides must be greater than the length of the remaining side), these distributions may still not fully mimic the constraints encountered in real-world scenarios.

---

> > > > > ### Author Response · Authors · 2024-11-24
> > > > > **rebuttal to Reviewer 4KBM part 5**
> > > > >
> > > > > > The physics equations in SRSD are made up of complex equations with few inputs (see the equations classified as 'Hard' in SRSD).
> > > > >
> > > > > The most challenging functions in SRSD often involve multiple variables, such as \(B_1, B_2\). The difficulty in discovering and accurately representing these functions typically arises from two primary factors: the scaling difficulty and the intrinsic complexity of the equation itself. While SRSD may address both aspects, we believe it is crucial to focus on equations that emphasize only one of these challenges. For example, we would focus on equations that demonstrate complexity in their structure, such as $\frac{\sqrt{-x_1^4 + 2x_1^2x_2^2 + 2x_1^2x_3^2 - x_2^4 + 2x_2^2x_3^2 - x_3^4}}{4}$, or those that illustrate scaling difficulties, such as $\frac{x_1+x_3+x_5}{2}$.
> > > > >
> > > > > Additionally, physical equations in SRSD often lack symmetry between variables, which can widely be discovered in nature.
> > > > >
> > > > > And  we emphasize that  that our benchmark complements SRBench, with a special focus on cases with ground truth solutions, making the recovery rate the only useful metric. A high recovery rate ensures high scores in $R^2$ or similar metrics like MSE, making the discussion of $R^2$ meaningless in this context. This is the most **fundamental** criterion we should really look into.
> > > > >
> > > > > And for real-world data, using $R^2$ to evaluate accuracy is also inappropriate. Metrics like normalized MSE or other alternatives are preferable to $R^2$.

---

> ### Author Response · Authors · 2024-11-25
> **Request your feedback before the end of the discussion period**
>
> Dear Reviewer 4KBM:
>
> As the author-reviewer discussion period will end soon, we would appreciate it if you could review our responses at your earliest convenience. If there are any further questions or comments, we will do our best to address them before the discussion period ends.
>
> Thank you very much for your time and efforts. Looking forward to your response!
>
> Sincerely,
>
> The Authors

---

### Official Review · Reviewer_jkuh · 2024-11-04

**Soundness:** 3
**Presentation:** 3
**Contribution:** 2
**Rating:** 6
**Confidence:** 5

**Summary:**

This paper contributes a benchmark of symbolic regression methods using synthetic problems derived from 2D and 3d geometry. It includes a description of these problems, a review of other benchmarking work in SR, and a benchmark of several SR methods on the proposed benchmark.

**Strengths:**

The paper is easy to follow and covers a lot of existing literature. The description of the problems is clear and for the most part the benchmarked methods cover a wide swatch of proposed algorithms. It may be of interest to those looking to benchmark new symbolic regression techniques.

**Weaknesses:**

While the paper has many strengths, one of the main weaknesses is that it does not make it clear to the reader why this set of synthetic benchmark problems is a good assessment of the efficacy of these methods in any real-world applications. The benchmark datasets are noiseless geometric functions, which do represent some aspect of the real-world, but this reviewer is left wondering how a real-world problem in which SR would be applied would relate to these problems. The authors do describe the significance of the equations mathematically, but do not offer a compelling example of why one would expect the best algorithms from this benchmark to perform well in some significant real-world version of these problems.

As an example, with other benchmarks like Feynman and Strogatz, we can imagine a scientist collecting the observations that comprise the dataset from real-world phenomena and using SR to discover the underlying physical relation governing the system. Even these problems suffer from a lack of realism which is why noise is synthetically added in prior benchmarks (SRBench) or data is collected from physical examplar systems (Schmidt & Lipson 2009). But what is the equivalent real-world setting for these geometry problems? And, how well can we expect this benchmark's assessment of methods to transfer to those tasks, especially when the data is noiseless?

Below are some more specific critiques:

> SR offers greater interpretability and superior generalization, avoiding the complexity often
associated with opaque models.

Two strong claims made without justification - I suggest softening the language or referencing examples.

L046-065: I found this to be a confusing and somewhat apocryphal description of SR history. One might argue that polynomial fitting and/or even classification and regression trees are all versions of symbolic regression, but the authors are stretching in refering to those methods as the origin of SR. SR as it is thought of today (structural and parametric equation fitting to data) is typically dated to Koza's 1992 book Genetic Programming. The paragraph on constant optimization is also not representative of this history; constant optimization for expression trees didn't begin with a "linear regression framework", they started as "ephemeral random constants" (ERCs), i.e. additional terminals made of random values that were "tuned" via expression tree manipulation. The integration of constant optimization into expression tree SR is often attributed to Topchy and Punch 2003. The authors also seem to describe stochastic hill climbing of constants which made popular in Lipson's lab's papers (maybe first with Bongard & Lipson PNAS 2005 or Zykov, Bongard & Lipson IEEE 2005) but they don't describe it clearly or cite it appropriately.

In their discussion of benchmarking history, the authors should also mention McDermott et al's 2012 paper, GP needs better benchmarks (http://dl.acm.org/citation.cfm?id=2330273), which was a bellwether moment for many in the SR community, who were often benchmarking on overly simplistic datasets. It was a key moment in this history as it incorporated community feedback and black-listed several overly simplistic SR benchmarks problems like the quartic polynomial and many of the Nguyen problems. Incidentally, as the field of SR has crossed to new communities, this step forward is often lost, and results in recent papers benchmarking on datasets that were deemed too easy many years prior.

The introduction description of SRBench only describes half of the datasets in the benchmark paper, and neglects to mention (importantly) that half of SRBench was dedicated to benchmarking on real-world datasets. These are crucial for benchmarking since otherwise SR methods are being compared only on synthetic data which comes with a load of limitations.

> Furthermore, the benchmark includes some scientifically unrealistic assumptions, such as treating the gravitational constant as a
variable, and has been criticized for its oversimplified sampling process and inappropriate formulas
(Matsubara et al., 2022).

The gravitational constant treatment is from the Feynman equations, and not specific to SRBench. Matsuraba et al's critique applied to the synthetic datasets in Feynman and ODE Strogatz, but the authors here attribute it to SRBench.

> DEAP (d’Ascoli et al., 2022)

This is the wrong reference; d'Ascoli et al 2022 is not about DEAP.

Furthermore, it is puzzling that the authors benchmark against DEAP and gplearn, which are both super simple and traditional GP-based SR approaches (they date from 2012 and 2016 but implement the algorithm as it was in the 90s), and then against PySR which is more recent. Why not pick other more recent GP-based SR method that have performed well in other benchmarking works, e.g. Operon (2020) etc?

Figure 4 should label the subplots as easy, medium, hard. If space allows it would be nice for this figure to be in the main text.

**Questions:**

Please see above for questions

---

> ### Author Response · Authors · 2024-11-23
> **rebuttal to Reviewer jkuh**
>
> **Q1:** Why does the benchmark focus on synthetic problems rather than real-world applications, which are typically noisy and complex?
>
> **A1:** Our dataset is designed to identify symbolic equations that are both simple and explainable to effectively solve problems. We have intentionally composed this dataset of ground truth equations rather than real-world scenarios that lack verifiable explanations for their functions. Thus, real-world datasets fall out of our scope.
>
> **Q2:** Why does the description only cover half of the datasets in SRBench, omitting those based on real-world data?
>
> **A2:** While the other part of SRBench includes real-world data, this study is focused on evaluating symbolic regression capabilities, making the rest of SRBench less relevant for our specific tests.
>
> **Q3:** Why benchmark against older, simpler methods instead of more recent, high-performing ones?
>
> **A3:** Operon is indeed a robust baseline, and we have included it in our tests to ensure a comprehensive evaluation.Full result can be found at global rebuttal or section F in revised paper.
>
> **Q4:** Can you address specific critiques more thoroughly?
>
> **A4:** We have revised the problematic sentences for clarity and accuracy.
>
> **Q5:** Why you choose geometric data to create dataset?
>
> **A5:** We choose geometric data for these reason:
>
> Patterns in Geometric: Unlike the Feynman dataset, which encompasses equations from diverse regions and subjects, our dataset is specifically focused on geometric data within a defined area. We concentrate on discovering patterns in geometric properties such as volume, area, and length. The primary motivation for selecting geometric equations is their inherent potential to unveil these patterns.
>
> Structured Learning Progression: The Feynman dataset includes a few sequences that progress from easy to difficult, such as the series from I.6.20 a to I.6.20 b. Our dataset, however, clearly illustrates many such progressions: for instance, from Helen's law to the calculation of circumcircle or incircle radii, which utilize Helen's law, or from Pythagoras' theorem to the cosine law, with the former being a special case of the latter. These process facilitates a deeper and more sequential learning experience.
>
>
>
> Realistic Constraints in Equations: Our dataset includes equations with generational constraints, such as the triangle constraint where the sum of two edges must exceed the third, and their difference must be less. These constraints make our data more realistic compared to data from SRbench, which is typically generated from uniform distributions. This approach ensures that our dataset not only supports the discovery of geometric relationships but also adheres more closely to real-world scenarios.
>
>
>
> Complex Equations with Few Inputs: Geometry excels at establishing intricate relationships between variables using a minimal number of tokens, as exemplified by Helen’s law. In symbolic regression, inputs are often chosen or crafted through feature engineering to reduce their number, but this does not necessarily simplify the underlying relationships between them. Therefore, having complex equations with few inputs is crucial because it challenges the models to uncover deep relationships without relying on a large number of variables.
>
> **Q6:** The history of symbolic regression is wrong.
>
> **A6:** We have revised the manuscript by relocating the entire paragraph on symbolic regression methods to Section 2. Consequently, the timeline presented in the introduction was incorrect. We have addressed this by eliminating the time references in the introduction, treating the discussion of the expression tree and constant optimization solely as background information.
>
> **Q7:** Incorrect references compared to baselines.
>
> **A7:** We have updated the paper to correct these issues.

---

> ### Author Response · Authors · 2024-11-25
> **Request your feedback before the end of the discussion period**
>
> Dear Reviewer jkuh:
>
> As the author-reviewer discussion period will end soon, we would appreciate it if you could review our responses at your earliest convenience. If there are any further questions or comments, we will do our best to address them before the discussion period ends.
>
> Thank you very much for your time and efforts. Looking forward to your response!
>
> Sincerely,
>
> The Authors

---

### Author Response · Authors · 2024-11-23
**Difference from SRbench**

We think we have 5 major different from the SRbench

Purpose of the Dataset: Our dataset is designed to identify symbolic equations that are both simple and explainable to effectively solve problems. We have intentionally composed this dataset of ground truth equations rather than real-world scenarios that lack verifiable explanations for their functions. Consequently, traditional metrics like R-squared or other error measures are not applicable to our goals since they do not align with our focus on explainability.



Patterns in Geometric: Unlike the Feynman dataset, which encompasses equations from diverse regions and subjects, our dataset is specifically focused on geometric data within a defined area. We concentrate on discovering patterns in geometric properties such as volume, area, and length. The primary motivation for selecting geometric equations is their inherent potential to unveil these patterns.



Structured Learning Progression: The Feynman dataset includes a few sequences that progress from easy to difficult, such as the series from I.6.20 a to I.6.20 b. Our dataset, however, clearly illustrates many such progressions: for instance, from Helen's law to the calculation of circumcircle or incircle radii, which utilize Helen's law, or from Pythagoras' theorem to the cosine law, with the former being a special case of the latter. These process facilitates a deeper and more sequential learning experience.



Realistic Constraints in Equations: Our dataset includes equations with generational constraints, such as the triangle constraint where the sum of two edges must exceed the third, and their difference must be less. These constraints make our data more realistic compared to data from SRbench, which is typically generated from uniform distributions. This approach ensures that our dataset not only supports the discovery of geometric relationships but also adheres more closely to real-world scenarios.



Complex Equations with Few Inputs: Geometry excels at establishing intricate relationships between variables using a minimal number of tokens, as exemplified by Helen’s law. In symbolic regression, inputs are often chosen or crafted through feature engineering to reduce their number, but this does not necessarily simplify the underlying relationships between them. Therefore, having complex equations with few inputs is crucial because it challenges the models to uncover deep relationships without relying on a large number of variables.



The results from our benchmark also differ from those of SRBench. Many baselines in SRBench focus primarily on the R² score, which may suggest they are better at fitting curves. However, their capability to accurately recover true symbolic equations is lacking. Moreover, thanks to the Structured Learning Progression, we are able to categorize these symbolic equations and assess model performance across different levels of difficulty. Additionally, the Patterns in Geometry enable us to evaluate each model's performance within specific patterns. This understanding allows us to select better baselines for future problem-solving involving these patterns.

---

### Author Response · Authors · 2024-11-23
**Extra experiment part1**

We have expanded our dataset to include four additional experiments concerning noise, speed, more baselines, and the introduction of constants.

Noise: We incorporated datasets with two levels of noise—1e-5 and 0.0001—to evaluate how well the models perform under noisy conditions. Unlike typical setups where only the target variable is affected by noise, we introduced noise equally to both the input variables and the target. This simulates a more realistic scenario where the measurement of both features and targets may be impacted by noise.

Speed: To assess the computational efficiency of each baseline, we measured the speed by averaging the results of 100 parallel runs for each category. Understanding the speed of each model is crucial as it allows us to create a Pareto front that balances the recovery rate against the computational time cost, providing a comprehensive view of model performance.

More Baselines: Recognizing the importance of robust comparison, we included additional baselines from the era of genetic programming. Specifically, we added Operon to our benchmarking table to evaluate its performance against other established methods.

Adding Constants: Since our dataset primarily comprises geometric equations where the only constant is π, we tested the ability of the symbolic regression models to handle constants by multiplying each dataset by a uniform constant ranging from 0 to 5. This test aims to assess each baseline's capability in accurately recovering symbolic expressions that incorporate constants.

The outcomes of these experiments are detailed below, illustrating how each model fares across these varied conditions and providing insights into their overall robustness and effectiveness in symbolic regression tasks.

 Table 1: result of data with noise> In each column, the noise level is before the slash and the metric is after that, the discount means the recovery rate discount between no noise with $10^{-3}$ noise.

| baselines | 0-count | 0-recovery | 0.00001-count | 0.00001-recovery | 0.001-count | 0.001-recovery | discount |
|---------|:-------:|:----------:|:-------------:|:----------------:|:-----------:|:--------------:|:--------:|
| PSRN|52 | 68.75% |47|59.65%|40 | 48.32% |70.29%|
| Pysr|48 | 59.31% |44|53.51%|44 | 49.31% |83.13%|
| NGGP|56 | 64.86% |50|52.37%|48 | 50.84% |78.39%|
| UDSR|46 | 47.11% |43|37.56%|43 | 35.56% |75.47%|
| RSRM|57 | 72.92% |52|61.15%|48 | 51.89% |71.17%|
| KAN |4|4.39% |4|3.78%|4|3.11% |70.68%|
| BMS |45 | 54.55% |39|44.74%|37 | 38.74% |71.01%|
|PhySO|25 | 24.44% |25|22.90%|23 | 21.81% |89.23%|
| symindy |26 | 29.18% |20|25.37%|18 | 21.34% |73.11%|
| gplearn |25 | 16.46% |23|13.19%|21 | 11.99% |72.83%|
| deap|37 | 22.65% |30|17.85%|27 | 17.54% |77.43%|
| EQL |3|0.94% |2|0.83%|2|0.67% |70.94%|
|Sindy|16 | 22.54% |15|19.35%|13 | 17.14% |76.05%|
| SPL |22 | 23.99% |20|21.55%|18 | 20.16% |84.03%|
| E2E |28 | 29.39% |23|25.64%|20 | 21.06% |71.64%|
| TPSR|24 | 24.01% |14|19.06%|10 | 14.85% |61.84%|
| AIF |23 | 25.13% |18|21.87%|17 | 15.60% |62.10%|
| SNIP|12 | 11.82% |10|10.23%|6|6.31% |53.36%|
| DGSR|35 | 43.17% |29|34.60%|28 | 30.41% |70.44%|
| NSRS|15 | 18.10% |9|15.70%|5| 10.42% |57.59%|

from the result, all baselines suffer a lot from noise. And baselines with transformer pre-train module like SNIP, NSRS suffers most. And AIF RSRM also does not perform well due to their searching algorithm is not able to displace these noise.

As well, PhySO and pysr still have low discount due to their symbolic ability on physical dimensions. With the dimension , they can cut a lot of useless equation. Also, dimension is not affected through noise.

Table 2:the average time cost within each baselines within 71 datasets and 100 parallel runs

| baselines | time cost(s) |
| --------- | :----------: |
| PSRN| 270|
| Pysr| 374|
| NGGP| 2341 |
| UDSR| 3512 |
| RSRM| 2794 |
| KAN | 130|
| BMS | 2371 |
| physo | 2098 |
| symindy | 478|
| gplearn | 523|
| deap| 476|
| EQL | 1209 |
| Sindy |0 |
| SPL | 1438 |
| E2E |1 |
| TPSR| 3602 |
| AIF | 3475 |
| SNIP| 2975 |
| DGSR| 1097 |
| NSRS| 746|

Next comes the speed test. In this test, we test each model's running speed through all 71 datasets. SIndy model, KAN model runs fast due tothey are linear model. Also, end2end transformer model alsoruns fast for it only runs once and optimize its constant.

Then is the new baselines. Operon can reach 50.1% with 45 reachable, which is sightly below PySR within 127s average, but much more better than gplearn or deap.

---

### Author Response · Authors · 2024-11-23
**extra experiment part 2**

Final is the constant learning:

 Table 3: result of data with constant, the discount means the recovery rate discount between no constantand with constant.

| baselines | nonconstant-recovery | constant-recovery | discount |
| --------- | -------------------- | ----------------- | -------- |
| PSRN| 68.75% | 51.21%| 74.49% |
| Pysr| 59.31% | 56.48%| 95.23% |
| NGGP| 64.86% | 59.78%| 92.17% |
| UDSR| 47.11% | 42.96%| 91.19% |
| RSRM| 72.92% | 63.98%| 87.74% |
| KAN | 4.39%| 3.97% | 90.49% |
| BMS | 54.55% | 47.24%| 86.60% |
| PhySO | 24.44% | 23.58%| 96.49% |
| symindy | 29.18% | 27.16%| 93.07% |
| gplearn | 16.46% | 15.82%| 96.09% |
| deap| 22.65% | 20.23%| 89.32% |
| EQL | 0.94%| 0.97% | 103.00%|
| Sindy | 22.54% | 22.00%| 97.60% |
| SPL | 23.99% | 22.45%| 93.56% |
| E2E | 29.39% | 26.13%| 88.91% |
| TPSR| 24.01% | 22.12%| 92.11% |
| AIF | 25.13% | 21.86%| 86.98% |
| SNIP| 11.82% | 10.85%| 91.81% |
| DGSR| 43.17% | 39.79%| 92.17% |
| NSRS| 18.10% | 16.84%| 93.05% |

We can conclude that PSRN can not good at handle with constant while the others can fit as well as before since the constant is only multiplies outside the equation.

---

### Meta-Review · Area_Chair_ZTDH · 2024-12-19

**Metareview:**

**(a) Summary**

This paper introduces a new benchmark for symbolic regression (SR) to address several limitations of the existing SRBench benchmark. The proposed benchmark includes 71 datasets derived from 2D and 3D geometric objects, with ground truth equations provided. The dataset is categorized into three difficulty levels, enabling a more structured benchmarking of SR methods. A total of 20 SR methods were evaluated on the proposed benchmark, with their performance assessed using the recovery rate metric.

**(b) Strengths**

Benchmarking for SR methods is still in its early stages compared to other well-established machine learning tasks, such as regression and classification. The addition of a new benchmark is a relevant and valuable contribution to the SR research community.

The presentation is clear, and the paper is easy to follow.

**(c) Weaknesses**

As highlighted by the reviewers, existing SR benchmark datasets, such SRSD (formerly known as the AI-Feynman database), already address benchmarking of symbolic regression to some extent. The advantages of the proposed benchmark over these existing datasets are fundamentally unclear. Specifically, the reviewers raised concerns regarding the lack of real-world datasets, the absence of datasets with diverse noise levels, and the lack of R2 score-based evaluation.

**(d) Decision Reasoning**

While I acknowledge the authors' efforts, the identified weaknesses are significant and need to be thoroughly addressed. A key issue with the current submission is the lack of a direct evaluation of the proposed benchmark dataset itself. Without addressing these concerns, the paper does not provide sufficient justification for the introduction of the new benchmark. Therefore, I recommend rejecting the paper.

**Additional Comments On Reviewer Discussion:**

Although some concerns were addressed in the author rebuttal, and I appreciate the revisions made to the paper based on the reviews, the improvements are still insufficient. At least one more round of a major revision is required before the paper can be considered for publication.

---

### Decision · Program_Chairs · 2025-01-22

Reject